# Caracal: Causal Architecture via Spectral Mixing

**Bingzheng Gan** [1]  **Tianyi Zhang** [1]  **Yusu Li** [1]  **Jing Huang** [1]  **Wei Shi** [1]  **Yangkai Ding** [1]  **Tao Yu** [1]

## Abstract

The scalability of Large Language Models to long sequences is hindered by the quadratic cost of attention and the limitations of positional encodings. To address these, we introduce **Caracal**, a novel architecture that replaces attention with a parameter-efficient, $\mathcal{O}(L \log L)$ Multi-Head Fourier (MHF) module. Our contributions are threefold: (1) We leverage the Fast Fourier Transform (FFT) for sequence mixing, inherently addressing both bottlenecks mentioned above. (2) We apply a frequency-domain causal masking technique that enforces autoregressive capabilities via asymmetric padding and truncation, overcoming a critical barrier for Fourier-based generative models. (3) Unlike efficient models relying on hardware-specific implementations (e.g., Mamba), we uses standard library operators. This ensures robust portability, eliminating common deployment barriers. Evaluations demonstrate that **Caracal** performs competitively with Transformer and SSM baselines, offering a scalable and simple pathway for efficient long-sequence modeling. Code is available in Appendix E.

## 1. Introduction

While the Transformer (Vaswani et al., 2017) remains the standard for sequence modeling, its reliance on attention introduces two structural bottlenecks. The first is computational complexity: operations scale quadratically with input sequence length $L$, i.e., $\mathcal{O}(L^2)$, making long sequences prohibitively expensive. The second is permutation equivariance; attention is insensitive to token order, necessitating external positional encodings. These encodings, whether absolute or relative, often tether performance to training context lengths, causing instability during extrapolation and imposing a context window limit.

One major line of research, the *Enhancement Paradigm*, aims to mitigate these issues by modifying the attention framework. To address the computational cost, sparse attention methods reduce complexity by introducing fixed connectivity patterns, as seen in Longformer (Beltagy et al., 2020) and BigBird (Zaheer et al., 2020), or by employing adaptive, content-based mechanisms like those in Reformer (Kitaev et al., 2020) and Routing Transformer (Roy et al., 2021). While effective, these methods risk information loss. Concurrently, sophisticated relative positional encodings like RoPE (Su et al., 2024), YaRN (Peng et al., 2024), and ALiBi (Press et al., 2022), which encode relative distances or apply distance-aware biases, have dramatically improved length extrapolation. However, the growing complexity of these techniques shows they are sophisticated compensations for a mechanism lacking a built-in sequence concept, motivating the search for a more fundamental alternative.

A more radical approach, the *Replacement Paradigm*, replaces the attention mechanism with more efficient alternatives. State Space Models (SSMs), particularly the recent Mamba architecture (Gu & Dao, 2023; Dao & Gu, 2024), have emerged as a powerful contender. Mamba achieves linear-time complexity and state-of-the-art performance, but its efficiency heavily relies on hardware-specific custom CUDA kernels, hindering portability, modification, and broad adoption. Another prominent direction within this paradigm leverages the Fourier Transform, offering an appealing $\mathcal{O}(L \log L)$ complexity for global token mixing (Lee-Thorp et al., 2022). However, these Fourier-based models have historically struggled with generative tasks. Their primary flaw lies in the difficulty of enforcing **causality**—a strict requirement for autoregressive decoding—within the frequency domain, a challenge that has relegated them to encoder-only or non-autoregressive applications.

In this work, we argue that a solution can be found that is both algorithmically elegant and hardware-agnostic. We introduce **Caracal**, a novel architecture that directly confronts the foundational issues of attention. By replacing the global attention mechanism with our **Multi-Head Fourier (MHF)** module, **Caracal** inherently resolves the quadratic complexity and the need for positional encodings. Furthermore, by applying a causal masking technique that operates in frequency domain, we overcome the critical causality barrier that has hindered previous Fourier-based models.

[1]Huawei Technologies Co., Ltd., China. Correspondence to: Bingzheng Gan <gbz1108@gmail.com>.

*Proceedings of the 43ʳᵈ International Conference on Machine Learning*, Seoul, South Korea. PMLR 306, 2026. Copyright 2026 by the author(s).

To maximize local feature extraction precision, we retain a small proportion of attention layers, constraining them to a sliding window mechanism. Crucially, this design preserves our architecture's fundamental advantages: the MHF layers inherently provide global positional information, eliminating the need for explicit positional encodings even in the attention layers, while the fixed-size sliding window of attention ensures the overall computational complexity remains efficient and scalable. Our contributions are as follows:

- **A Novel Autoregressive Fourier Module:** We propose the Multi-Head Fourier (MHF) module, which mixes token information via a gated element-wise product in the frequency domain, directly replacing the attention layer in a Transformer.

- **Frequency-Domain Causal Masking:** We apply a technique that enforces causality entirely in the frequency domain using asymmetric padding and truncation. This addresses a key obstacle that has hindered the application of Fourier models to generative tasks.

- **A Position-Encoding-Free Architecture:** Our model obviates the need for explicit positional encodings because the Fourier Transform inherently captures position through its sinusoidal basis functions. This design is theoretically advantageous for length extrapolation.

- **Solid Validation:** We show that **Caracal** achieves performance competitive with state-of-the-art Transformer and SSM baselines on diverse benchmarks. While its $\mathcal{O}(L \log L)$ complexity is above the $\mathcal{O}(L)$ scaling of SSMs, it remains significantly more efficient than the $\mathcal{O}(L^2)$ of Transformers, all while bypassing the hardware-dependent optimization hurdles and implementational complexities typically inherent to SSMs.

**Conflict of Interest Disclosure**: The authors are employed by Huawei, which leads the development of **Caracal**, which was among the ones evaluated in this paper.

## 2. Related Work

To address the limitations of Transformers, researchers have sought efficient alternatives, primarily categorized into State Space Models (SSMs) rooted in continuous control theory, Linear Recurrent Models that optimize hidden state update rules for associative memory, and Spectral Methods which leverage the efficiency of the Fast Fourier Transform (FFT).

### 2.1. State Space Models (SSMs)

SSMs have recently emerged as a powerful efficient replacement for attention. Foundational work like the S4 (Gu et al., 2022b) and S4D (Gu et al., 2022a) demonstrates the potential for modeling long-range dependencies by leveraging a formulation computable as a global convolution. However, the time-invariant nature of S4's state matrices limits its ability to perform content-aware reasoning.

To address this, subsequent architectures introduce gating mechanisms and data-dependent primitives. H3 (Fu et al., 2023b) introduces multiplicative gating to mimic linear attention. Hyena (Poli et al., 2023) employs implicit long convolutions by generating filters via a small MLP acting on positional encodings, leveraging FFTs for sub-quadratic processing. While effective, Hyena's filters are strictly position-based (dependent on $t$), whereas **Caracal**'s filters are content-based (generated dynamically from input tokens $x$), enabling direct semantic interactions akin to attention.

Most recently, Mamba (Gu & Dao, 2023) and Mamba-2 (Dao & Gu, 2024) achieve state-of-the-art performance via selective, input-dependent state updates. While improving expressivity, this selectivity **sacrifices the parallel convolutional form** inherent in earlier SSMs, necessitating complex hardware-aware scans or specialized block-partitioning to maintain efficiency. Such SSD-style models often involve a high conceptual barrier, requiring deep expertise in state-space duality and structured matrix decomposition. This mathematical complexity can lead to a "black-box" nature that makes it challenging to modify internal logic without risking architectural breakdowns. Furthermore, achieving peak performance in modern SSMs often requires manual tuning of block sizes or reliance on custom CUDA kernels, hindering portability. In contrast, **Caracal** provides superior structural clarity and flexibility by leveraging standard FFT operators, ensuring universal compatibility and ease of modification as further detailed in Section 3.3.

### 2.2. Linear Recurrence and Associative Memory

Another trajectory optimizes the recurrent update rules of hidden states. RetNet (Sun et al., 2023) uses a dual form for parallel training and $\mathcal{O}(1)$ inference, while GLA (Yang et al., 2024a) adds hardware-efficient gating. DeltaNet (Yang et al., 2024b) and Gated DeltaNet (Yang et al., 2025) employ a (gated) delta rule to selectively update hidden states, a concept further formalized via associative memory with momentum-based updates (Behrouz et al., 2025). Alternatively, TTT (Sun et al., 2025) frames modeling as test-time training using self-supervised updates. While effective at state compression, these models often require complex update rules or specialized kernels. **Caracal** bypasses explicit iterative updates, achieving dynamic, content-aware mixing through the simplicity of global convolution.

### 2.3. Fourier and Spectral Methods

Another line of research leverages the Fourier Transform to achieve global token mixing with $\mathcal{O}(L \log L)$ complexity.

### 2.3.1. SPECTRAL METHODS AS ENHANCEMENTS

This category integrates Fourier-based operators into existing architectures for efficiency. Both Fourier Transformer (He et al., 2023) and FwNet-ECA (Mian et al., 2025) insert a Fourier block after attention to compress representations, while Vim-F (Zhang et al., 2024) augments a Mamba block with a parallel Fourier filtering module. Similarly, FAN (Dong et al., 2024) replaces MLP activations with trigonometric functions. Other works use spectral methods to assist the attention mechanism. FSAT (Zhuang et al., 2022) predicts sparse masks via Fourier convolution, and FourierNAT (Kiruluta et al., 2025) employs a Fourier-mixing block to aid non-autoregressive decoding. While demonstrating the utility of spectral biases, these methods do not fundamentally remove the attention bottleneck.

### 2.3.2. SPECTRAL METHODS AS REPLACEMENTS

More ambitious approaches use Fourier-based modules to completely substitute attention. The seminal FNet (Lee-Thorp et al., 2022) replaces attention with a parameter-free 2D FFT in BERT. This inspires vision models like GF-Net (Rao et al., 2021) and AFNO (Guibas et al., 2022), building on FNO (Li et al., 2021), as well as DCT-Former (Scribano et al., 2023) which uses discrete cosine transforms. However, these models are inherently non-causal and restricted to encoder-only tasks.

Recent attempts to adapt spectral methods for autoregressive generation face structural rigidity. SPECTRE (Fein-Ashley et al., 2025) relies on fixed sliding windows, fragmenting long-range dependencies. Similarly, FlashButterfly (Fu et al., 2023c) explicitly parameterizes a static global kernel; fixed at training time, it lacks inherent length extrapolation capabilities. In contrast, **Caracal** generates filters dynamically from the input via local operations, naturally adapting to arbitrary sequence lengths.

Finally, regarding hardware efficiency, Monarch Mixer (Fu et al., 2023a) approximates convolutions via GEMMs to maximize utilization, whereas **Caracal** prioritizes algorithmic simplicity and portability through standard FFT operators. Therefore, a truly causal, dynamic, and implementationally simple Fourier-based replacement for attention remains an open challenge—one that **Caracal** addresses.

## 3. Methodology

### 3.1. Overall Structure

The design of **Caracal** prioritizes simplicity and efficiency. We take a standard decoder-only Transformer, such as GPT-2 (Radford et al., 2019), as a blueprint and apply minimal modifications. As illustrated in Figure 1, a **Caracal** model is structurally very similar to a Transformer model, with two key differences:

1. The global Masked Multi-Head Self-Attention modules are replaced by our **Multi-Head Fourier (MHF)** modules. We retain a small number of attention layers, restricted to a sliding window mechanism, to capture local dependencies without quadratic cost.

2. The **Positional Encoding** module is removed. The MHF module inherently captures global positional information and propagates this context to subsequent layers (including the retained attention blocks), rendering explicit positional encodings redundant.

This design allows **Caracal** to easily leverage mature components from the existing Transformer ecosystem, such as the feed-forward network, layer normalization, and residual connections.

### 3.2. Multi-Head Fourier (MHF) Module

The MHF module is the heart of **Caracal**, responsible for performing global, causal information mixing in $\mathcal{O}(L \log L)$ complexity. Given an input $x \in \mathbb{R}^{B \times L \times D}$, where $B$ is the batch size, $L$ is the sequence length, and $D$ is the model dimension, the forward pass proceeds as follows:

**Step 1: Injecting Local Inductive Bias.** We pass the input $x$ through a lightweight, depthwise causal 1D convolution to capture local syntactic patterns (e.g., n-grams):

$$\tilde{x} = \text{CausalConv1d}(x) \tag{1}$$

This convolution uses a small kernel (e.g., $k = 3$) with left-sided padding to ensure causality. This step compensates for the removal of explicit positional encodings, allowing the subsequent global Fourier Transform to focus on longer-range semantic dependencies.

**Step 2: Preparing Gated Signals.** We first apply Layer Normalization, obtaining $x_{\text{norm}} = \text{LayerNorm}(\tilde{x})$. We then project this into parallel content $x_v$ and gate $x_g$ streams. The content stream is generated via a linear projection operator:

$$x_v = \text{Linear}_V(x_{\text{norm}}) \tag{2}$$

The gate stream $x_g$ involves nested operators to enable intra-head communication:

$$x_g = \text{Conv1d}_{G2}(\sigma(\text{Linear}_{G1}(x_{\text{norm}}))) \tag{3}$$

where $\sigma$ is SiLU and $\text{Conv1d}_{G2}$ is 1D group convolution (kernel size 1) with $n_{\text{head}}$ groups. This configuration facilitates intra-head interaction, allowing channels within each head to collectively learn a shared gating representation.

**Step 3: Causal Mixing in Frequency Domain.** Convolution in the time domain equals multiplication in the frequency domain. To enforce causality, we pad sequences to

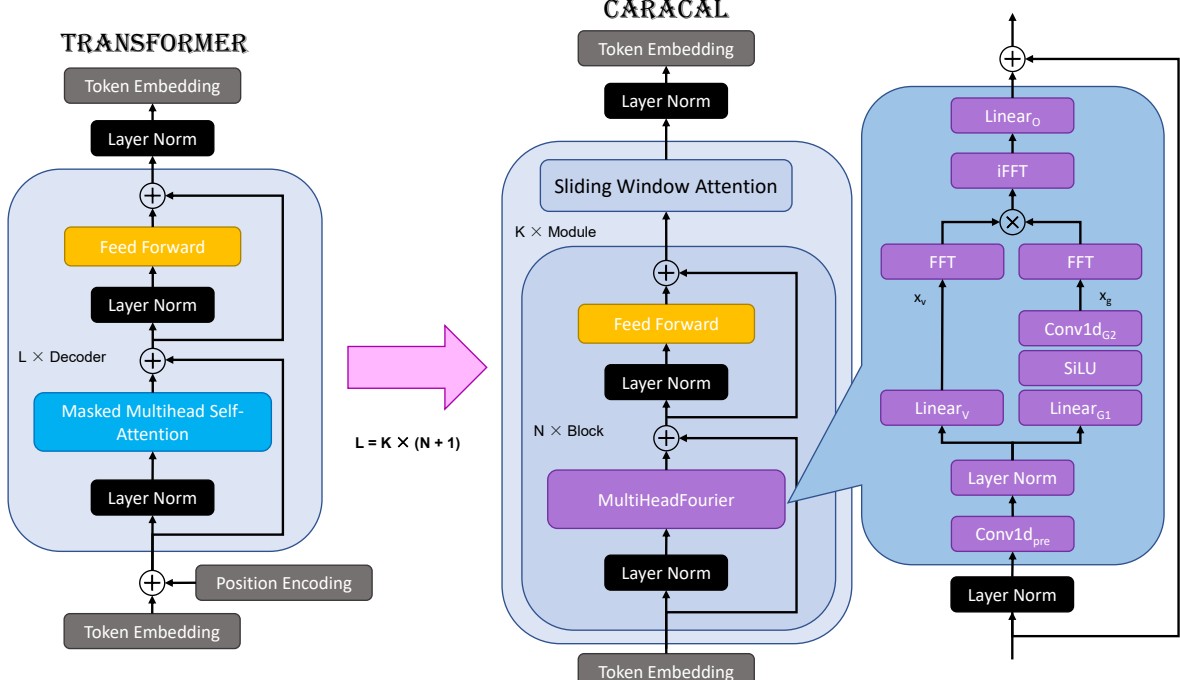

*Figure 1.* Comparison of a standard Transformer model (left) and our proposed **Caracal** model (right). The core modification is the replacement of Masked Multi-Head Attention with our Multi-Head Fourier (MHF) module and the removal of positional encodings.

double length ($N = 2L$) and transform them via the FFT:

$$V_{\text{fft}} = \mathcal{F}(\text{Pad}(x_v)), \quad G_{\text{fft}} = \mathcal{F}(\text{Pad}(x_g)) \tag{4}$$

In this expanded frequency space, we perform element-wise multiplication:

$$X_{\text{fft}} = V_{\text{fft}} \odot G_{\text{fft}} \tag{5}$$

During this spectral mixing stage, all operations are applied channel-independently along the sequence length $L$. There is no further interaction across heads or channels, where $G_{\text{fft}}$ adaptively modulates $V_{\text{fft}}$ frequencies based on context.

**Step 4: Causal Reconstruction and Projection.** We transform the mixed spectrum back to the time domain using the inverse FFT and truncate the result to the original length $L$ to remove the non-causal padding artifacts:

$$x_{\text{mixed}} = \text{Truncate}(\mathcal{F}^{-1}(X_{\text{fft}})) \tag{6}$$

This "pad-FFT-multiply-iFFT-truncate" pipeline is mathematically equivalent to a causal convolution. Finally, a linear projection integrates the mixed information:

$$y = \text{Linear}_O(x_{\text{mixed}}) \tag{7}$$

To provide a concrete overview of the data flow, we present the forward pass pseudocode in Appendix A.

### 3.3. Architectural Properties

**Computational Complexity:** The cost of MHF stems from FFT and iFFT, each with a complexity of $\mathcal{O}(L \log L)$, a significant improvement over attention's $\mathcal{O}(L^2)$.

**Implicit Positional Information:** The basis functions of the Fourier Transform inherently contain ordered frequency and phase information. Hence, the model can implicitly perceive token positions without external positional encodings.

**Structural Clarity and Flexibility:** Unlike selective SSMs relying on custom CUDA kernels, **Caracal** uses a standard causal convolutional form. This design offers significantly greater flexibility, allowing researchers to transparently innovate on components—such as the pre-convolution or gating mechanisms—while leveraging standard, highly-optimized FFT libraries. This ensures high performance by default and universal compatibility across diverse environments without the need for hardware-specific manual tuning.

## 4. Motivation

In this section, we delve deeper into the architectural principles of **Caracal**, providing a first-principles comparison to attention, analyzing how our design overcomes the challenge in prior spectral models, and showing how our method of data-dependent mixing fits in with the SSMs.

### 4.1. A First-Principles View: Attention and FFT as Weighted Sums

From first principles, both Attention and the Fourier Transform can be viewed as mechanisms for token mixing via a weighted sum. This fundamental similarity provides the theoretical grounding for why one can replace the other.

In a single head of attention, the updated representation for the $t$-th token, $r_t$, is a weighted sum of the value vectors $v_j$ from all L tokens in the context:

$$r_t = \sum_{j=0}^{L-1} \alpha_{tj} v_j \tag{8}$$

Here, the weights $\alpha_{tj}$ are computed dynamically based on the query of the $t$-th token and the key of the $j$-th token ($\alpha_{tj} = \text{softmax}(q_t \cdot k_j^T / \sqrt{d_k})$. These weights are data-dependent, forming a dense $L \times L$ attention matrix.

We deconstruct the Discrete Fourier Transform (DFT) from this perspective. For a sequence of vectors $v_0, v_1, \ldots, v_{L-1}$ (analyzed dimension-wise), the DFT produces a sequence of vectors $r_0, r_1, \ldots, r_{L-1}$. According to the definition of DFT, the standard formula for the $t$-th output vector is:

$$r_t = \sum_{j=0}^{L-1} v_j \cdot e^{-i \frac{2\pi}{L} tj} = \sum_{j=0}^{L-1} w_{tj} v_j \tag{9}$$

Eqn. 9 shows the Fourier Transform parallels attention: both mix tokens globally via weighted sums. However, a critical distinction lies in the weights. While attention weights $\alpha_{tj}$ are data-dependent, Fourier weights $w_{tj}$ are static and determined solely by relative positions. While efficient, this static nature limits dynamic flexibility. **Caracal** compensates for this rigidity by employing instance-specific kernels that bridge the performance gap while retaining the efficiency of the FFT (see Appendix B).

### 4.2. The Causality Dilemma in Autoregressive Spectral Models

Previous attempts to replace attention with FFTs struggle in generative models due to the challenge of enforcing causality efficiently. Autoregressive models must ensure the prediction for token $t$ depends only on tokens $0, \ldots, t$. Attention solves this via a causal mask. The key is that the attention weights $\alpha_{tj}$ are explicitly computed as an intermediate matrix **before** the final weighted sum is performed (Eqn. 8). This creates a crucial window of opportunity to intervene: the mask forcibly sets all weights where $j > t$ to zero, ensuring future tokens do not contribute to the output for token $t$. This is all performed in a single, parallel forward pass, allowing for highly efficient training.

A naive application of the FFT does not afford this luxury. The Fast Fourier Transform is a highly optimized algorithm

that computes the full weighted sum (Eqn. 9) directly from the input sequence. **There is no intermediate step where a modifiable matrix of explicit weights $w_{tj}$ is formed. One cannot apply a mask to the Fourier basis functions 'midway' through the computation; the algorithm yields the final sum in what is essentially a single, atomic operation.**

Since the weights themselves cannot be masked, the only way to enforce causality is to manipulate the input sequence. To generate the correct output for token $t$, one must feed the FFT algorithm only the causal portion of the sequence $(v_0, \ldots, v_t)$ while hiding the portion after token $t$ $(v_{t+1}, \ldots, v_{L-1})$. This leads to a significant computational inefficiency during training. To get the outputs for all $L$ tokens in a sequence, one would need to perform $L$ separate FFT computations of increasing length. The total complexity for a single training step would approach $\mathcal{O}(L^2 \log L)$, which is significantly less efficient than the single pass of attention $\mathcal{O}(L^2)$. This computational dilemma is the fundamental reason why most prior spectral models are confined to non-causal encoders or processed inputs in fixed-length, non-causal chunks.

**Caracal** resolves this dilemma by leveraging the mathematical equivalence between frequency-domain multiplication and time-domain causal convolution (see Appendix C). Instead of attempting to mask the Fourier basis functions directly, we utilize the well-established "pad-FFT-multiply-iFFT-truncate" procedure to compute a causal output in a single, parallel forward pass. This approach treats the spectral workflow as a computationally efficient realization of the following underlying target calculation for each token $t$:

$$r_t = \sum_{j=0}^{t} v_j g_{t-j} \tag{10}$$

where $v_j$ denotes the $j$-th temporal slice of the projected feature tensor $x_v$ (Eqn. 2) along the sequence dimension (specifically $x_v[:, :, j, :]$ in the implementation), where $x_v$ and $x_g$ share a shape of [B, H, L, d] (see Appendix B). The term $g_{t-j}$ is defined analogously as the $(t-j)$-th slice of the tensor $x_g$ (Eqn. 3). By reformulating the spectral operation as this discrete causal summation, **Caracal** ensures that the prediction for any token $t$ remains strictly independent of future information with an index larger than $t$. This allows the model to overcome the bottleneck that hindered previous spectral models, maintaining the causal rigor required for generative tasks without sacrificing training efficiency.

### 4.3. Achieving Data-Dependent Mixing with Algorithmic Efficiency

A key factor in the performance of modern sequence models is their ability to perform data-dependent or content-aware reasoning. The evolution from S4 to Mamba illustrates this principle. S4, with its time-invariant state matrices, is

algorithmically efficient and computable as a parallel convolution. However, its data-independent nature limits its expressive power. Mamba's core innovation is to introduce selective, input-dependent state updates, dramatically improving performance. This improvement, however, comes at the cost of breaking the parallel convolutional structure, necessitating a hardware-aware sequential scan algorithm that relies on custom, low-level optimizations.

**Caracal** offers a direct path to achieving data-dependent mixing while preserving algorithmic efficiency. In our architecture, the effective "convolutional kernel" is the gate stream $x_g$, which is generated dynamically from the input $x$ via a small neural network (Linear $\rightarrow$ SiLU $\rightarrow$ Conv1d). This makes the mixing operation fully data-dependent.

Crucially, because this interaction is formulated as a multiplication in the frequency domain, it remains equivalent to a convolution in the time domain. This formulation preserves the globally parallel computational structure that is inherent to convolutions and FFTs. Consequently, **Caracal** benefits from the expressive power of data-dependent mixing, much like Mamba, but without sacrificing the hardware independence and universal parallelism of standard library operators. It achieves content-aware reasoning through a purely algorithmic and elegant mechanism.

# 5. Experiments

To thoroughly validate **Caracal**, we conduct extensive experiments. We first evaluate effectiveness and scaling capabilities across various benchmarks. Next, we quantify training efficiency in terms of throughput across increasing context lengths. Finally, we provide an in-depth parameter study and ablation study to justify our architectural design.

## 5.1. Experimental Setup

**Baselines:** We compare **Caracal** against:

- **Llama:** A standard Transformer architecture (Touvron et al., 2023) using RoPE (Su et al., 2024) and SwiGLU (Shazeer, 2020).
- **Mamba & Mamba-2:** State-of-the-art pure SSMs (Gu & Dao, 2023; Dao & Gu, 2024).
- **Jamba:** A hybrid architecture (Lenz et al., 2025) interleaving Mamba layers with Attention, serving as a direct competitor to our hybrid spectral design.

We scale all models from "Tiny" to "Large". The configurations are chosen to align with the GPT-3 paper (Brown et al., 2020), ensuring a fair comparison of parameter efficiency. The hyperparameters are detailed in Table 1. For **Caracal**, we insert a Sliding Window Attention (SWA) layer (window size 256) after every two MHF layers (ratio 2:1).

*Table 1.* Hyperparameter configurations for the different model sizes used in our experiments.

| MODEL SIZE | $d_{\text{MODEL}}$ | $n_{\text{LAYER}}$ | $n_{\text{HEAD}}$ | $d_{\text{HEAD}}$ |
|---|---|---|---|---|
| TINY (T) | 512 | 12 | 8 | 64 |
| SMALL (S) | 768 | 12 | 12 | 64 |
| MEDIUM (M) | 1024 | 24 | 16 | 64 |
| LARGE (L) | 1536 | 24 | 16 | 96 |

To ensure a high-performance and fair comparison across all experiments, we employ hardware-optimized kernels for all models. Specifically, Mamba and Mamba-2 utilize the library `mamba_ssm` for Selective Scan and SSD operators. The Llama baseline is implemented via the library `transformers`, leveraging the native FlashAttention implementation provided by PyTorch's scaled dot product attention (SDPA). Correspondingly, the SWA layers in our hybrid **Caracal** are accelerated using the flash attention operator to ensure competitive throughput.

**Configurations:** We pre-train all models from scratch on the **FineWeb-10B** dataset (Penedo et al., 2024), a high-quality corpus of English web text. This dataset is a filtered subset of Common Crawl and has become a standard resource for training foundation models of this scale. All models are trained for 10B tokens with a context length of 512. We employ the AdamW optimizer with $\beta_1 = 0.9$, $\beta_2 = 0.95$, and a weight decay of 0.1. The training utilizes a global batch size of 0.5M tokens. The learning rate follows a cosine schedule, peaking at $9 \times 10^{-4}$ after a linear warmup over the first 3.75% of training steps. We also apply gradient clipping with a norm of 1.0. The `gpt2` tokenizer is used throughout all training and evaluation stages. All models are trained on a single node equipped with eight consumer-grade GPUs, each featuring 24GB of VRAM.

**Benchmarks:** We evaluate on a diverse suite of tasks using `lm-evaluation-harness`:

- **Common Sense Reasoning:** Hellaswag (Zellers et al., 2019), Winogrande (Sakaguchi et al., 2020), ARC-E/C (Clark et al., 2018), PIQA (Bisk et al., 2020), SIQA (Sap et al., 2019), BoolQ (Clark et al., 2019).
- **Language Modeling:** LAMBADA (Paperno et al., 2016) (reporting both Accuracy and Perplexity).
- **Long-Context Understanding:** SWDE (Lockard et al., 2019), FDA (Arora et al., 2023) (retrieval & extraction over extended sequences).

## 5.2. Effectiveness and Scaling

### 5.2.1. GENERAL PERFORMANCE AND SCALABILITY

We first evaluate the effectiveness of **Caracal** on standard language modeling and common sense reasoning bench-

*Table 2.* Effectiveness and Scaling. We report accuracy (%) for all tasks and ppl for Lambada. Bold numbers indicate the best performance within each parameter class. Values in parentheses denote the total parameter count for each model.

| Size | Model(Params) | LMB. PPL ↓ | LMB. ACC ↑ | Hella. ACC ↑ | ARC-e ACC ↑ | ARC-c ACC ↑ | Wino. ACC ↑ | BoolQ ACC ↑ | PIQA ACC ↑ | SIQA ACC ↑ | Avg. ACC ↑ |
|---|---|---|---|---|---|---|---|---|---|---|---|
| T | Llama(64M) | 164.19 | **22.53** | 30.97 | 45.16 | 24.91 | **50.59** | 55.84 | 60.83 | 36.13 | 40.87 |
|   | Mamba(66M) | **129.88** | 20.36 | **31.87** | 45.88 | 24.32 | 48.70 | 58.69 | 61.81 | 37.31 | 41.12 |
|   | Mamba2(64M) | 191.20 | 17.62 | 31.36 | **46.17** | 24.15 | 49.72 | 55.93 | 60.88 | 36.28 | 40.26 |
|   | Jamba(71M) | 158.41 | 21.99 | 31.29 | 46.13 | 24.57 | 50.20 | 52.97 | **61.97** | **37.41** | 40.82 |
|   | **Caracal(63M)** | 219.90 | 20.51 | 30.46 | 44.91 | **26.02** | 50.28 | **59.30** | 61.64 | 36.03 | **41.14** |
| S | Llama(124M) | 79.94 | **28.06** | 34.26 | 47.98 | 25.34 | **51.93** | 56.70 | 62.95 | 36.90 | 43.02 |
|   | Mamba(129M) | 86.33 | 25.25 | **34.99** | **49.87** | 25.60 | 50.20 | **60.67** | 63.87 | **38.38** | **43.60** |
|   | Mamba2(125M) | 100.76 | 22.36 | 34.50 | 48.23 | 25.00 | 50.91 | 59.79 | 63.11 | 37.21 | 42.64 |
|   | Jamba(138M) | **60.48** | 27.50 | 34.43 | 48.65 | 26.02 | 51.85 | 59.14 | 62.79 | 37.56 | 43.49 |
|   | **Caracal(120M)** | 92.05 | 25.07 | 33.65 | 47.69 | **26.96** | 51.38 | 59.88 | **64.69** | 37.46 | 43.35 |
| M | Llama(360M) | **32.65** | **34.06** | 41.21 | 53.37 | **29.35** | 52.41 | 60.55 | 66.65 | 38.95 | **47.07** |
|   | Mamba(372M) | 45.72 | 28.51 | **41.86** | **54.17** | 29.27 | 50.83 | 60.98 | 66.32 | **39.15** | 46.39 |
|   | Mamba2(357M) | 51.97 | 29.38 | 41.17 | 53.75 | 29.18 | 50.99 | 60.86 | 66.65 | 38.79 | 46.35 |
|   | Jamba(409M) | 42.44 | 33.09 | 41.62 | 52.95 | 27.73 | 52.17 | 60.00 | 66.05 | 38.69 | 46.54 |
|   | **Caracal(345M)** | 38.50 | 32.25 | 39.89 | 51.81 | 28.07 | 52.25 | **61.35** | **67.68** | 38.49 | 46.47 |
| L | Llama(757M) | **24.92** | **36.74** | 44.97 | 55.22 | 29.44 | 52.64 | 61.47 | 69.21 | **40.12** | 48.73 |
|   | Mamba(793M) | 34.34 | 34.02 | **46.02** | 59.97 | 31.23 | 51.93 | **62.11** | 67.03 | 39.66 | 49.00 |
|   | Mamba2(764M) | 36.32 | 33.18 | 45.65 | **60.61** | 29.27 | 52.64 | 61.83 | 67.46 | 39.36 | 48.75 |
|   | Jamba(866M) | 26.93 | 36.35 | 45.87 | 56.90 | **31.40** | 52.57 | 61.31 | 68.99 | 39.20 | **49.07** |
|   | **Caracal(724M)** | 29.39 | 35.26 | 45.10 | 58.16 | 29.69 | **53.20** | 61.90 | **69.26** | 39.51 | 49.01 |

*Table 3.* Comparison with Broader Baselines. To ensure a fair comparison with results reported in Behrouz et al. (2025), we evaluate **Caracal (Default)** and its variant **Caracal w/o SWA** (excluding SWA layers) using their protocol (345M parameters, 15B tokens, 4096 context length). Accuracy (%) and ppl for Lambada are reported; bold numbers indicate the best performance.

| Model | LMB. PPL ↓ | LMB. ACC ↑ | Hella. ACC ↑ | ARC-e ACC ↑ | ARC-c ACC ↑ | Wino. ACC ↑ | BoolQ ACC ↑ | PIQA ACC ↑ | SIQA ACC ↑ | Avg. ACC ↑ |
|---|---|---|---|---|---|---|---|---|---|---|
| Transformer++ | 41.08 | 30.76 | 34.76 | 45.21 | 24.05 | 50.53 | 58.24 | 62.98 | 36.81 | 42.92 |
| RetNet | 49.73 | 28.24 | 34.15 | 44.27 | 23.62 | 50.91 | 59.72 | 62.61 | 36.79 | 42.54 |
| GLA | 43.02 | 28.73 | 35.96 | 54.19 | 24.29 | 50.00 | 58.39 | 64.05 | 37.13 | 44.09 |
| Mamba | 40.21 | 29.94 | 35.88 | 49.24 | 24.56 | 49.82 | 60.07 | 63.79 | 35.41 | 43.59 |
| DeltaNet | 47.30 | 28.43 | 35.95 | 52.68 | 25.37 | 49.63 | 58.79 | 63.52 | 37.96 | 44.04 |
| TTT | 34.19 | 30.06 | 35.71 | 53.01 | 26.11 | 50.08 | 59.83 | 63.97 | 37.32 | 44.51 |
| Gated DeltaNet | 30.94 | 34.11 | 38.12 | 55.28 | 26.77 | 51.60 | 59.54 | 63.08 | 34.89 | 45.42 |
| Moneta | 29.31 | **35.70** | 39.23 | **55.96** | 27.15 | 52.04 | 60.22 | 63.99 | 37.29 | 46.45 |
| Yaad | **29.11** | 34.09 | 39.86 | 54.75 | **28.64** | 51.12 | 60.29 | 64.93 | 33.82 | 45.94 |
| Memora | 30.44 | 33.68 | 39.17 | 53.40 | 27.99 | 51.23 | 59.29 | 65.21 | 34.10 | 45.51 |
| **Caracal w/o SWA** | 56.43 | 25.79 | 38.54 | 50.42 | 26.54 | 52.88 | 60.00 | 66.32 | 38.84 | 44.92 |
| **Caracal (Default)** | 37.27 | 32.70 | **40.95** | 50.46 | 27.90 | **53.12** | 60.80 | **67.95** | 39.10 | **46.62** |

marks, focusing on its scaling behavior and its standing among diverse architectural paradigms.

**Direct Scaling Comparison:** Table 2 presents the results of **Caracal** scaled from "Tiny" to "Large" sizes under a strictly controlled training setting (10B tokens, context length 512). The primary goal here is to establish a clear scaling trajectory for our model. The results indicate that **Caracal** exhibits consistent and predictable performance gains as parameter counts increase. Across various benchmarks, **Caracal** achieves performance that is competitive with pure Transformer (Llama), pure SSM (Mamba/Mamba2) and hybrid architecture (Jamba) baselines.

**Broad Architectural Benchmarking:** To situate **Caracal** within a wider ecosystem and evaluate its performance under extended context training without re-training every sub-quadratic baseline, we adopt the configuration from Behrouz et al. (2025). By aligning with their recipe (340M parameters, 15B tokens, 4096 context length), we can directly compare **Caracal** against a diverse suite of reported baselines. As shown in Table 3, **Caracal (Default)** achieves state-of-the-art performance, while **Caracal w/o SWA** maintains a competitive ranking. This highlights the significant gains from SWA layers while demonstrating the robust capability of our core MHF module.

*Table 4.* Evaluation on Information Extraction and Retrieval over extended sequences. Models are compared at the "Large" size. We report accuracy (%) for SWDE and FDA benchmarks.

| MODEL (L) | SWDE ↑ | FDA ↑ | AVG. ↑ |
|---|---|---|---|
| LLAMA | **16.38** | **2.72** | **9.55** |
| MAMBA | 8.91 | 2.00 | 5.46 |
| MAMBA2 | 8.55 | 1.81 | 5.18 |
| JAMBA | 9.54 | 2.27 | 5.91 |
| **CARACAL** | 8.82 | 1.91 | 5.37 |

### 5.2.2. INFORMATION EXTRACTION ON LONG CONTEXT

In Table 4, we evaluate **Caracal** on tasks requiring information extraction and retrieval over extended sequences (SWDE and FDA). The experimental results indicate that while attention-based model (e.g., Llama) maintains a distinct performance advantage in these scenarios, **Caracal** exhibits retrieval capabilities on par with other modern sub-quadratic architectures. This positions **Caracal** as a competitive and implementationally simpler alternative for modeling long-range dependencies.

The performance gap between **Caracal** and Llama highlights a fundamental architectural trade-off: while the dense pointwise matching in attention enables near-perfect fine-grained retrieval, **Caracal**—similar to other SSM models—prioritizes global computational efficiency, which results in lower "resolution" for exact extraction. However, **Caracal** offers a superior Pareto frontier for global sequence modeling, providing high scalability while remaining performant across various general benchmarks. We consider the current architecture a robust baseline and will explore methods to bridge this retrieval gap in future work.

### 5.3. Efficiency and Speed

A central motivation for the design of **Caracal** is to achieve high computational efficiency, especially as the industry shifts toward ultra-long sequences. To evaluate this, we measure the total training time and token throughput for the "Tiny" size of **Caracal** and various competitive baselines. In addition to our default hybrid configuration, we also evaluate a "pure" variant (**Caracal w/o SWA**) to isolate the specific overhead of the MHF. The "Tiny" configuration is specifically selected to ensure that even the longest sequences could fit within the 24GB memory limit of our hardware. As detailed in Table 2, all models have approximately 65M parameters and are trained using the identical hardware and hyperparameter configurations described in Section 5.1.

Figure 2 illustrates the training time and throughput across varying context lengths $L$ (Detailed numbers see Appendix D). While the attention-based Llama architecture exhibits the expected quadratic complexity $\mathcal{O}(L^2)$, resulting in a

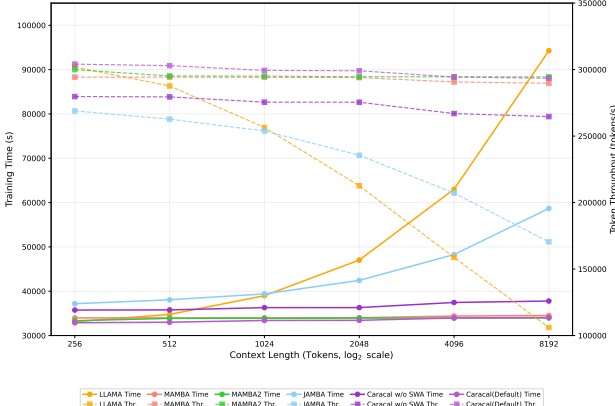

*Figure 2.* Efficiency comparison across varying context lengths $L \in \{256, \ldots, 8192\}$. The primary y-axis (left) denotes total training time, while the secondary y-axis (right) shows token throughput. **Caracal** exhibits a near-linear scaling profile, maintaining high throughput even at $L = 8192$.

sharp degradation in throughput and a surge in training time as $L$ scales, **Caracal** maintains a significantly more favorable $\mathcal{O}(L \log L)$ scaling profile.

As the context length increases, the efficiency advantage of **Caracal** becomes increasingly pronounced. At a context length of 8192, **Caracal** is nearly **three times as fast as** Llama in terms of total training time. Notably, our empirical results show that the computational efficiency of **Caracal** is **comparable** to that of strictly linear-time $\mathcal{O}(L)$ models such as Mamba and Mamba2. The other hybrid architectures like Jamba, as expected, exhibit a performance profile that sits between the highly efficient SSMs and the standard Transformer. Interestingly, the hybrid **Caracal** exhibits slightly higher throughput than its pure MHF counterpart. This is due to the integration of SWA layers with a window size of 256, which benefits from the highly optimized FlashAttention kernels. While MHF provides superior long-range scaling, the hardware-level acceleration of SWA reduces the constant-factor overhead for those specific layers.

These results demonstrate that **Caracal** effectively mitigates the quadratic bottleneck of traditional attention. It offers a highly performant and accessible alternative that matches the efficiency of state-space models. Simultaneously, **Caracal** retains the robust global modeling capabilities inherent to Fourier-based information integration.

### 5.4. Ablation Study

We evaluate the contributions of the core components in **Caracal** through ablation studies. All tests are performed using the "Large" size model configuration across our evaluation suite in Table 2. We examine the impact of Sliding-Window Attention (**SWA**), Pre-Convolution (**PC**, the short 1D causal convolution in Step 1, Section 3.2), Positional

*Table 5.* Ablation study variants. We report the *Avg. Acc* (%), representing the mean accuracy across the benchmarks listed in Table 2. All experiments were conducted using the "Large" model.

| MODEL VARIANT | AVG. ACC (%) |
|---|---|
| FULL MODEL (DEFAULT) | **49.01** |
| W/O SWA | 48.22 |
| W/O SWA & PC | 47.82 |
| WITH PE | 48.94 |
| SSLP | 48.05 |

*Table 6.* Parameter study on the hybridization ratio of MHF to SWA layers. We report the *Avg. Acc* (%), representing the mean accuracy across the benchmarks listed in Table 2. All experiments were conducted using the "Small" (12-layer) model.

| RATIO (MHF:SWA) | AVG. ACC (%) |
|---|---|
| 5:1 | 43.04 |
| 3:1 | 43.19 |
| 2:1 (DEFAULT) | **43.35** |
| 1:1 | 43.26 |

Encodings (**PE**), and our two-stage gated design. The model variants evaluated include:

- **Full Model**: The Standard **Caracal** configuration as described in Section 5.1, featuring interleaved SWA layers and PC in MHF.

- **w/o SWA**: A "Pure MHF" variant where all SWA layers are replaced by MHF layers.

- **w/o SWA & PC**: A variant that further removes PC layer in MHF.

- **with PE**: The full model with additional explicit PE to test the necessity of auxiliary positional signals.

- **SSLP**: Replaces the two-stage gated design (Linear$_{G1}$ and Conv1d$_{G2}$) with a Single-Stage Linear Projection.

Table 5 reveals: **1) Local Priors:** Local inductive biases from SWA and PC are crucial for fine-grained linguistic consistency, though pure MHF remains competitive, proving the robustness of the spectral mixing mechanism. **2) Positional Awareness:** Explicit PE provides no significant gain, confirming that MHF's dynamic Toeplitz structure inherently captures relative positions, making auxiliary positional signals redundant. **3) Gating Complexity:** Our two-stage design outperforms the SSLP variant. This hierarchical approach enables richer intra-head communication and the development of more expressive, context-aware filters.

### 5.5. Parameter Study

To identify the optimal integration of Fourier-domain global mixing and local attention refinement, we conduct a param-

eter study on the hybridization ratio of Multi-Head Fourier (MHF) layers to Sliding Window Attention (SWA) layers. Using the "Small" size (12 layers), we adjust the frequency of SWA layers within the model backbone. To maintain a balanced structure, we follow a uniform interleaving strategy: a single SWA layer is inserted after every $k$ blocks of MHF layers, ensuring that local refinement is distributed consistently across all representation levels.

Table 6 shows the impact of these ratios on the model's overall effectiveness across our evaluation suite in Table 2. Starting from a sparse integration (5:1), we observe a steady improvement in accuracy as more SWA layers are introduced, peaking at the **2:1 hybridization ratio**.

Further increasing the attention density to a 1:1 ratio slightly degrades performance, suggesting that while SWA provides vital local features, **Caracal**'s strength lies in the global mixing of MHF layers. Over-relying on local attention disrupts the synergy between the frequency-domain global modeling and the time-domain local extraction.

Thus, our default **2:1 ratio** is the empirical "sweet spot" for **Caracal**, maximizing the collaborative potential of both modules. This allows MHF layers to efficiently build a global foundation while SWA layers provide the local precision needed for peak performance.

## 6. Conclusion

In this paper, we introduced **Caracal**, a sequence modeling architecture designed to address the computational and context-scaling bottlenecks inherent in modern Large Language Models. By replacing attention with a data-dependent Fourier-based gated mixing module, **Caracal** achieves a complexity of $\mathcal{O}(L \log L)$, offering a more scalable alternative for processing extended sequences. Through our frequency-domain causal masking technique, we successfully adapt spectral methods for autoregressive generation, overcoming the structural constraints that previously limited their utility in generative tasks.

Our design prioritizes simplicity and hardware portability by relying exclusively on standard, optimized operators. This approach ensures high architectural flexibility, allowing researchers to refine the model's internal logic without the complexity of low-level programming. Empirical results indicate that **Caracal** exhibits robust scaling and maintains performance competitive with mainstream Transformer and SSM models. While localized retrieval remains an area for further refinement, **Caracal** establishes a practical and efficient foundation that enhances the feasibility of long-context modeling. We believe this work opens a promising path toward more accessible and scalable next-generation sequence models. A detailed discussion of limitations is provided in Appendix F.

## Impact Statement

This paper presents work whose goal is to advance the field of machine learning by introducing a more computationally efficient sequence modeling architecture. By replacing the quadratic complexity of attention with the log-linear complexity of the Fourier Transform, our approach **Caracal** has the potential to significantly reduce the energy consumption and carbon footprint associated with training and deploying large language models.

While fundamental architectures can be used for a wide range of applications, including those with potential for misuse (e.g., generating misinformation), our work does not introduce new inherent risks beyond those already present in existing large language models. We believe the efficiency gains contribute positively to the democratization of AI research by lowering the hardware barrier for long-context modeling. There are no other specific ethical consequences of our work that we feel must be highlighted here.

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

---

**Algorithm 1** The Multi-Head Fourier (MHF) Module Forward Pass

---

**Input**: Input sequence $x \in \mathbb{R}^{B \times L \times D}$
**Parameters**: All model weights $\theta$
**Output**: Output sequence $y \in \mathbb{R}^{B \times L \times D}$

 1: **Inject local inductive bias with causal convolution. Left-padding (pad=2) for causality.**
 2: $x \leftarrow \text{Conv1d}(x, \text{kernel\_size} = 3, \text{groups} = D)$
 3: $x_{\text{norm}} \leftarrow \text{LayerNorm}(x)$
 4: **Prepare content stream ($x_{\mathbf{v}}$) and gate stream ($x_{\mathbf{g}}$).**
 5: $x_{\text{v}} \leftarrow \text{Linear}(x_{\text{norm}})$
 6: $x_{\text{g}} \leftarrow \text{Conv1d}(\text{SiLU}(\text{Linear}(x_{\text{norm}})), \text{kernel\_size} = 1, \text{groups} = H)$
 7: **Perform causal mixing in frequency domain.**
 8: $N \leftarrow 2 \times L$         *Set padded length for causal convolution via FFT*
 9: $V_{\text{fft}} \leftarrow \text{RFFT}(x_{\text{v}}, \text{n} = N, \dim = 1)$         *Pad and transform content stream*
 10: $G_{\text{fft}} \leftarrow \text{RFFT}(x_{\text{g}}, \text{n} = N, \dim = 1)$         *Pad and transform gate stream*
 11: $X_{\text{fft}} \leftarrow V_{\text{fft}} \odot G_{\text{fft}}$         *Element-wise product in frequency domain*
 12: $x_{\text{mixed}} \leftarrow \text{IRFFT}(X_{\text{fft}}, \text{n} = N, \dim = 1)$         *Transform mixed spectrum back to sequence domain*
 13: **Truncate to enforce causality and project to output**
 14: $x_{\text{causal}} \leftarrow x_{\text{mixed}}[:, : L, :]$         *Truncate to original sequence length $L$*
 15: $y \leftarrow \text{Linear}(x_{\text{causal}})$
 16: **return** $y$

---

## A. Algorithm

To clarify the operational flow of our proposed Multi-Head Fourier (MHF) module, we present its forward pass in Algorithm 1. This algorithm details how causal convolution is integrated with frequency-domain mixing to model sequence data.

In Algorithm 1, the notation follows standard deep learning conventions. We denote the batch size as $B$, the input sequence length as $L$, and the model's hidden dimension as $D$. The number of parallel heads within the MHF module is denoted by $H$, where the dimension of each head is $d = D/H$.

A key implementation detail involves the `Conv1d` layers used in lines 2 and 6. Standard deep learning libraries typically expect 1D convolutional inputs in the format $(B, C, L)$, where $B$ is the batch size, $C$ is the number of channels, and $L$ is the sequence length. Therefore, to ensure the convolution is applied along the sequence length dimension, we first permute the input tensor from its shape $\mathbb{R}^{B \times L \times D}$ to $\mathbb{R}^{B \times D \times L}$. Following the convolution, the output is permuted back to $\mathbb{R}^{B \times L \times D}$ to ensure dimensional consistency for subsequent operations.

## B. Theoretical Analysis of Multi-Head Fourier (MHF) Expressivity

### B.1. Formal Definition and Dynamic Representational Capacity

Following the notation in Eqn. 8, we represent the Multi-Head Fourier (MHF) operation as a dynamic weighted sum. The tensors $\mathbf{x}_g, \mathbf{x}_v \in \mathbb{R}^{B \times H \times L \times d}$ are the outputs of the gating and value branches respectively (as defined in Section 3.2), where $B$ is the batch size, $H$ is the number of heads, $L$ is the sequence length, and $d$ is the head dimension.

For each batch $b$, head $h$, feature $c$, the MHF module computes the output $r_t$ at time $t$ via a channel-wise causal convolution:

$$r_t^{(b,h,c)} = \sum_{j=0}^{t} \alpha_{tj}^{(b,h,c)} v_j^{(b,h,c)}, \quad \text{where } \alpha_{tj}^{(b,h,c)} = x_g[b, h, t-j, c] \text{ and } v_j^{(b,h,c)} = x_v[b, h, j, c] \tag{11}$$

In this formulation, the gate stream $\mathbf{x}_g$ serves directly as the **time-domain filter coefficients**. Specifically, for any fixed channel $(b, h, c)$, the temporal sequence $x_g[b, h, 0 : L, c]$ defines an *instance-specific* convolution kernel. Unlike the weights $w_{tj}$ in a standard FFT (Eqn. 9) or the kernels in static CNNs that are fixed after training, the weights $\alpha_{tj}$ in MHF are dynamically generated from the input $\mathbf{x}$, enabling the model to adapt its filtering characteristics to the local context.

This adaptability can be formally analyzed through the lens of the **Volterra Series**, a functional expansion for non-linear

dynamical systems with memory. A causal functional mapping an input sequence to an output can be decomposed into kernels of increasing order:

$$r_t = \int h_1(\tau)v(t-\tau)d\tau + \int \int h_2(\tau_1, \tau_2)v(t-\tau_1)v(t-\tau_2)d\tau_1 d\tau_2 + \cdots + \int \cdots \int h_n(\tau_1, \ldots, \tau_n) \prod_{i=1}^{n} v(t-\tau_i)d\tau_1 \ldots d\tau_i$$

(12)

While a single MHF layer (Eqn. 11) realizes a first-order Volterra system where $\mathbf{x}_g$ act as the kernel $h_1$, its data-dependency allows the model to capture complex high-order interactions similar to how attention approximates sequence functions.

### B.2. Structural Constraints and Inductive Bias

The fundamental difference between MHF and Attention lies in the structural constraint on the weight matrix $\mathbf{A} = (\alpha_{tj})$. In Attention, $\mathbf{A}_{attn} \in \mathbb{R}^{L \times L}$ is a dense lower triangular matrix where each $\alpha_{tj}$ is computed independently. In contrast, MHF constrains $\mathbf{A}_{mhf}$ to be a **Lower Triangular Toeplitz Matrix** for each $(b, h, c)$ channel, where the weights satisfy $\alpha_{t,j} = \alpha_{t+1,j+1} = x_g[b, h, t-j, c] = x_g^{(b,h,c)}[t-j]$:

$$\mathbf{A}_{attn} = \begin{pmatrix} \alpha_{0,0} & 0 & \ldots & 0 \\ \alpha_{1,0} & \alpha_{1,1} & \ldots & 0 \\ \vdots & \vdots & \ddots & \vdots \\ \alpha_{L-1,0} & \alpha_{L-1,1} & \ldots & \alpha_{L-1,L-1} \end{pmatrix}, \quad \mathbf{A}_{mhf} = \begin{pmatrix} x_g^{(b,h,c)}[0] & 0 & \ldots & 0 \\ x_g^{(b,h,c)}[1] & x_g^{(b,h,c)}[0] & \ldots & 0 \\ \vdots & \vdots & \ddots & \vdots \\ x_g^{(b,h,c)}[L-1] & x_g^{(b,h,c)}[L-2] & \ldots & x_g^{(b,h,c)}[0] \end{pmatrix}$$

(13)

Consequently, while the number of free parameters (degrees of freedom) in $\mathbf{A}_{attn}$ is $\mathcal{O}(L^2)$, $\mathbf{A}_{mhf}$ possesses only $\mathcal{O}(L)$ degrees of freedom per sequence. This $\mathcal{O}(L)$ constraint is not a limitation but a strategic **Shift-Invariant** inductive bias.

**Theorem 1 (Efficiency of Symmetry).** *To represent a global shift $r_t = v_{t-1}$, Attention requires $\mathcal{O}(L^2)$ complexity to align $L$ query-key pairs. MHF represents this with a single parameter in the spectral domain.*

**Proof of Theorem 1:**

**1. Attention Representation:** To represent a unit shift $r_t = v_{t-1}$ for $t \in \{1, \ldots, L-1\}$, the Attention matrix $\mathbf{A}_{attn}$ must satisfy the following condition for each element:

$$\alpha_{t,j} = \delta(j - (t-1)) = \begin{cases} 1, & j = t-1 \\ 0, & \text{otherwise} \end{cases}$$

(14)

In a standard Attention mechanism, each row $t$ is computed as $\text{softmax}(\mathbf{q}_t \mathbf{K}^T / \sqrt{d})$. To achieve this shift pattern, the model must learn $L$ distinct query-key alignments such that for every $t$, the attention score peaks exclusively at position $t - 1$. Since each row $\mathbf{A}_{t,:}$ is functionally independent, the degrees of freedom required to specify this alignment across the full sequence length $L$ scale as $\mathcal{O}(L^2)$ within the matrix structure.

**2. MHF Representation (Toeplitz):** In MHF, the matrix $\mathbf{A}_{mhf}$ is constrained to be Toeplitz, defined by a kernel $x_g^{(b,h,c)}$. The shift $r_t^{(b,h,c)} = x_v^{(b,h,c)}[t-1]$ is represented by setting the kernel values as follows:

$$x_g^{(b,h,c)}(i) = \delta(i-1) = \begin{cases} 1, & i = 1 \\ 0, & \text{otherwise} \end{cases}$$

(15)

Because the Toeplitz structure dictates that $\alpha_{t,j} = x_g^{(b,h,c)}[t-j]$, setting the single value $x_g^{(b,h,c)}[1] = 1$ automatically ensures that $\alpha_{t,t-1} = 1$ for all $t \in \{1, \ldots, L-1\}$ simultaneously. This demonstrates that the shift-invariant inductive bias reduces the required degrees of freedom from $\mathcal{O}(L^2)$ to $\mathcal{O}(1)$ for such patterns.

**Conclusion:** The MHF hypothesis space is thus optimized for capturing periodicities and relative dependencies. By learning a global kernel $x_g^{(b,h,c)} \in \mathbb{R}^L$ through FFT, MHF achieves a global receptive field identical to attention but with a more efficient parameterization for structural patterns.

### B.3. Analysis of Proximity Bias and Alternative Formulations

A distinct characteristic of **MHF** is its departure from the explicit proximity inductive bias (recency bias) commonly found in State Space Models (SSMs) or Sliding Window Attention (SWA). To justify this design choice, we analyze the alternative "z-shifted cross-correlation" suggested during the peer-review process:

$$r_t^{(b,h,c)} = \sum_{j=0}^{t} v_j^{(b,h,c)} g_{j-t+z}^{(b,h,c)} \tag{16}$$

where $z$ acts as a fixed shift. Under this formulation (e.g., for $z = 2$), the interaction for token $t$ would be:

$$r_t^{(b,h,c)} = v_{t-2}^{(b,h,c)} g_0^{(b,h,c)} + v_{t-1}^{(b,h,c)} g_1^{(b,h,c)} + v_t^{(b,h,c)} g_2^{(b,h,c)} \tag{17}$$

While this ensures that $v_t^{(b,h,c)}$ interacts with a fixed "anchor" in the kernel, it inherently imposes a **local window constraint**. Specifically, as the sequence index $t$ increases, early tokens (where $j < t - z$) naturally shift out of the kernel's active range ($g$ index $< 0$), causing their influence to vanish.

In contrast, the standard causal convolution in MHF enables **global sequence mixing**. By setting $g_0^{(b,h,c)}$ as the consistent weight for the "current step" $v_t^{(b,h,c)}$, our model allows every historical token $v_j^{(b,h,c)}$ to maintain a continuous interaction with the kernel across the entire sequence length. As shown below:

- $r_0^{(b,h,c)} = \mathbf{v_0^{(b,h,c)}} \mathbf{g_0^{(b,h,c)}}$
- $r_1^{(b,h,c)} = v_0^{(b,h,c)} g_1^{(b,h,c)} + \mathbf{v_1^{(b,h,c)}} \mathbf{g_0^{(b,h,c)}}$
- $r_2^{(b,h,c)} = v_0^{(b,h,c)} g_2^{(b,h,c)} + v_1^{(b,h,c)} g_1^{(b,h,c)} + \mathbf{v_2^{(b,h,c)}} \mathbf{g_0^{(b,h,c)}}$
- $r_t^{(b,h,c)} = v_0^{(b,h,c)} g_t^{(b,h,c)} + v_1^{(b,h,c)} g_{t-1}^{(b,h,c)} + \cdots + v_{t-1}^{(b,h,c)} g_1^{(b,h,c)} + \mathbf{v_t^{(b,h,c)}} \mathbf{g_0^{(b,h,c)}}$

This mechanism allows $g_0^{(b,h,c)}$ to learn the immediate representational transformation for the current token, while $g_{dist}^{(b,h,c)}$ learns the decay or importance of past tokens at distance $dist$. This flexibility allows MHF to dynamically learn whether to emphasize recency or long-range context based on data, providing a more expressive alternative to fixed proximity priors.

## C. Formal Derivation of Causal Convolution via FFT Padding

To provide transparency regarding the "pad-FFT-multiply-iFFT-truncate" pipeline, we present a derivation based on the duality of **polynomial representations**. A polynomial $P_v(z) = \sum v_i z^i$ can be represented in two forms: the **Coefficient Form**, where the sequence $v = [v_0, v_1, \dots]$ represents the coefficients, and the **Point-Value Form**, where a sequence of evaluated points $\{P_v(z_i)\}$ uniquely determines the polynomial.

### a. The Goal: Causal Coefficient Multiplication

Direct linear convolution is identical to multiplying two polynomials to obtain the resulting coefficients. For a sequence length $L = 2$:

$$(v_0 + v_1 z) \cdot (g_0 + g_1 z) = \underbrace{(v_0 g_0)}_{r_0} + \underbrace{(v_1 g_0 + v_0 g_1)}_{r_1} z + \underbrace{(v_1 g_1)}_{r_2} z^2 \tag{18}$$

The first two coefficients represent the **causal outputs**: $r_0$ depends only on $v_0, g_0$, and $r_1$ depends only on information up to index 1. This satisfies the strict definition of causality in sequence modeling.

### b. The FFT "Shortcut": Form Conversion

Instead of the $\mathcal{O}(L^2)$ direct multiplication, the FFT-iFFT method achieves the same result via representation change:

**FFT (Evaluation):** It converts $v$ and $g$ from **Coefficient Form** into **Point-Value Form** by evaluating the polynomials $P_v(z)$ and $P_g(z)$ at $N$ roots of unity $\{z_i\}$.

**Point-wise Multiplication:** In this domain, the product's point-values are obtained by simple element-wise multiplication:

$$P_{out}(z_i) = P_v(z_i) \cdot P_g(z_i) \tag{19}$$

**iFFT (Interpolation):** The iFFT interpolates these product values $\{P_{out}(z_i)\}$ back into the **Coefficient Form** sequence $r$.

**c. Why Padding is Mandatory for Causality ($L = 2$ Example)**

The "circularity" of the FFT arises because it samples the polynomial at $N$ points where $z^N = 1$. This implies any term $z^N$ is treated as $z^0$ (modulo $z^N - 1$), causing higher-degree terms to "wrap around" to lower-degree positions.

**Without Padding ($N = L = 2$):** The FFT uses points where $z^2 = 1$. Consequently, the term $(v_1 g_1)z^2$ from the product wraps around to the $z^0$ position ($v_1 g_1 \cdot 1$). The resulting 0-th coefficient of the iFFT output $r'$ becomes:

$$r'_0 = v_0 g_0 + \mathbf{v_1 g_1} \tag{20}$$

Here, the "future" token $v_1$ influences the "past" output $r'_0$, violating causality.

**With Padding ($N = 2L = 4$):** We pad $v$ and $g$ to length 4 with zeros, denoted as $v'$ and $g'$. The FFT now uses points where $z^4 = 1$. In this case, $z^2$ is distinct from $z^0$ (only $z^4, z^8 \dots$ would wrap to $z^0$). The first two terms of the iFFT result $r'$ become:

- $r'_0 = v'_0 g'_0 + (\text{wrap-around terms}) = v_0 g_0 + 0 = \mathbf{r_0}$

- $r'_1 = v'_1 g'_0 + v'_0 g'_1 + (\text{wrap-around terms}) = v_1 g_0 + v_0 g_1 + 0 = \mathbf{r_1}$

**Conclusion:** As shown, padding ensures that the circular wrap-around does not contaminate the causal window $[0, L-1]$. By truncating the result back to length $L$, we strictly recover the causal coefficients $r_t$ by discarding the non-causal components (the tail of the linear convolution) in the range $[L, N-1]$.

## D. Detailed Efficiency Benchmarks

This section provides the exact numerical data summarized in Figure 2 to facilitate a more granular comparison of architectural efficiency. We report two primary metrics across varying context lengths $L \in \{256, \dots, 8192\}$. Table 7 measures the cumulative wall-clock time (seconds) required for a fixed training budget, serving as a proxy for total computational overhead and energy consumption. Table 8 records the number of tokens processed per second, directly reflecting the model's operational speed and its ability to scale to long sequences without the quadratic slowdown typical of standard attention.

*Table 7.* Total Training Time (seconds) across different context lengths.

| Model | 256 | 512 | 1024 | 2048 | 4096 | 8192 |
|---|---|---|---|---|---|---|
| Llama | 33,118 | 34,758 | 38,987 | 47,027 | 63,003 | 94,258 |
| Mamba | 33,975 | 33,977 | 33,994 | 34,014 | 34,398 | 34,516 |
| Mamba2 | 33,339 | 33,874 | 33,889 | 33,941 | 33,949 | 33,956 |
| Jamba | 37,184 | 38,064 | 39,393 | 42,441 | 48,267 | 58,673 |
| **Caracal** w/o SWA | 35,749 | 35,781 | 36,297 | 36,303 | 37,461 | 37,785 |
| **Caracal** (Default) | 32,883 | 33,005 | 33,401 | 33,431 | 33,965 | 34,071 |

*Table 8.* Token Throughput (tokens/s) across different context lengths.

| Model | 256 | 512 | 1024 | 2048 | 4096 | 8192 |
|---|---|---|---|---|---|---|
| Llama | 301,950 | 287,703 | 256,495 | 212,643 | 158,722 | 106,091 |
| Mamba | 294,334 | 294,316 | 294,169 | 293,996 | 290,714 | 289,720 |
| Mamba2 | 299,949 | 295,211 | 295,081 | 294,628 | 294,559 | 294,498 |
| Jamba | 268,932 | 262,715 | 253,852 | 235,621 | 207,180 | 170,436 |
| **Caracal** w/o SWA | 279,728 | 279,477 | 275,504 | 275,459 | 266,944 | 264,655 |
| **Caracal** (Default) | 304,108 | 302,984 | 299,392 | 299,123 | 294,420 | 293,504 |

# E. Complete Architecture Implementation

We provide the complete PyTorch implementation of the **Caracal** architecture below. This implementation includes the core Multi-Head Fourier (MHF) module, the hybrid Sliding Window Attention (SWA) layer, and the overall model structure.

```python
import torch
import torch.nn as nn
from torch.nn import functional as F
from flash_attn import flash_attn_func

class MultiHeadFourier(nn.Module):
    def __init__(self, d_model, n_heads):
        super().__init__()
        self.d_model = d_model
        self.n_heads = n_heads
        self.d_head = int(d_model / n_heads)
        self.pre_conv = nn.Conv1d(in_channels=self.d_model, out_channels=self.d_model,
            kernel_size=3, groups=self.d_model, bias=False)
        self.ln = nn.LayerNorm(self.d_model)
        self.silu = nn.SiLU()
        self.W_V = nn.Linear(in_features=self.d_model, out_features=self.d_model)
        self.W_G1 = nn.Linear(in_features=self.d_model, out_features=self.d_model)
        self.W_G2 = nn.Conv1d(in_channels=self.d_model, out_channels=self.d_model,
            kernel_size=1, groups=self.n_heads)
        self.linear = nn.Linear(in_features=self.d_model, out_features=self.d_model)
        self.linear.NEED_SCALE_INIT = 1

    def forward(self, x):
        # x: [batch_size, seq_len, d_model]
        x_permuted = x.permute(0, 2, 1)
        # x_permuted: [batch_size, d_model, seq_len]
        padding = self.pre_conv.kernel_size[0] - 1
        # padding = 2
        x_padded = F.pad(x_permuted, (padding, 0))
        # padded_x: [batch_size, d_model, seq_len+2]
        x = self.pre_conv(x_padded).permute(0, 2, 1)
        # x: [batch_size, seq_len, d_model]
        x_norm = self.ln(x)
        # x_norm: [batch_size, seq_len, d_model]
        batch_size, seq_len = x_norm.size(0), x_norm.size(1)
        N = 2 * seq_len
        x_v = self.W_V(x_norm).reshape(batch_size, seq_len, self.n_heads, self.d_head).\
            transpose(1,2)
        # x_v: [batch_size, n_heads, seq_len, d_head]
        x_g = self.W_G1(x_norm).transpose(1,2)
        # x_g: [batch_size, d_model, seq_len]
        x_g = self.W_G2(self.silu(x_g)).transpose(1,2)
        # x_g: [batch_size, seq_len, d_model]
        x_g = x_g.reshape(batch_size, seq_len, self.n_heads, self.d_head).transpose(1,2)
        # x_g: [batch_size, n_heads, seq_len, d_head]
        G_fft = torch.fft.rfft(x_g.to(torch.float32), n=N, dim=2)
        V_fft = torch.fft.rfft(x_v.to(torch.float32), n=N, dim=2)
        # G_fft: [batch_size, n_heads, N//2+1, d_head]
        # V_fft: [batch_size, n_heads, N//2+1, d_head]
        X_fft = G_fft * V_fft
        # X_fft: [batch_size, n_heads, N//2+1, d_head]
        x_fft = torch.fft.irfft(X_fft, n=N, dim=2)
        # x_fft: [batch_size, n_heads, N, d_head]
        x_fft = x_fft[:, :, :seq_len, :]
        # x_fft: [batch_size, n_heads, seq_len, d_head]
        x_fft = x_fft.transpose(1, 2).contiguous().reshape(batch_size, seq_len, self.\
            d_model)
        # x_fft: [batch_size, seq_len, d_model]
        x = self.linear(x_fft)
```

```
57          # x: [batch_size, seq_len, d_model]
58          return x
59
60
61  class SlidingWindowAttention(nn.Module):
62      def __init__(self, d_model, n_heads, window_size):
63          super().__init__()
64          self.d_model = d_model
65          self.n_heads = n_heads
66          self.window_size = window_size
67          self.d_head = d_model // n_heads
68          self.c_attn = nn.Linear(d_model, 3 * d_model, bias=False)
69          self.c_proj = nn.Linear(d_model, d_model, bias=False)
70          self.c_proj.NEED_SCALE_INIT = 1
71
72      def forward(self, x):
73          # x: [batch_size, seq_len, d_model]
74          batch_size, seq_len, _ = x.size()
75          qkv = self.c_attn(x)
76          # qkv: [batch_size, seq_len, 3*d_model]
77          qkv = qkv.view(batch_size, seq_len, self.n_heads, 3, self.d_head)
78          # qkv: [batch_size, seq_len, n_heads, 3, d_head]
79          dtype_original = qkv.dtype
80          qkv_hp = qkv.to(torch.bfloat16)
81          q = qkv_hp[:, :, :, 0]
82          k = qkv_hp[:, :, :, 1]
83          v = qkv_hp[:, :, :, 2]
84          # q,k,v: [batch_size, seq_len, n_heads, d_head]
85          y_hp = flash_attn_func(
86              q, k, v,
87              dropout_p=0.0,
88              softmax_scale=None,
89              causal=True,
90              window_size=(self.window_size - 1, 0)
91          )
92          y = y_hp.to(dtype_original)
93          # y: [batch_size, seq_len, n_heads, d_head]
94          y = y.reshape(batch_size, seq_len, self.d_model)
95          # y: [batch_size, seq_len, d_model]
96          return self.c_proj(y)
97
98
99  class MLP(nn.Module):
100     def __init__(self, d_model, intermediate_size):
101         super().__init__()
102         self.fc_1 = nn.Linear(in_features=d_model, out_features=intermediate_size, bias=
                False)
103         self.fc_gate = nn.Linear(in_features=d_model, out_features=intermediate_size,
                bias=False)
104         self.fc_2 = nn.Linear(in_features=intermediate_size, out_features=d_model, bias=
                False)
105         self.silu = nn.SiLU()
106         self.fc_2.NEED_SCALE_INIT = 1
107
108     def forward(self, x):
109         # x: [batch_size, seq_len, d_model]
110         x_g = self.silu(self.fc_gate(x))
111         # x_g: [batch_size, seq_len, intermediate_size]
112         x_v = self.fc_1(x)
113         # x_v: [batch_size, seq_len, intermediate_size]
114         x = x_v * x_g
115         # x: [batch_size, seq_len, intermediate_size]
116         x = self.fc_2(x)
117         # x: [batch_size, seq_len, d_model]
118         return x
```

```
119
120
121  class Block(nn.Module):
122      def __init__(self, d_model, n_heads, intermediate_size, window_size, layer_type):
123          super().__init__()
124          self.d_model = d_model
125          self.n_heads = n_heads
126          self.intermediate_size = intermediate_size
127          self.window_size = window_size
128          self.ln_1 = nn.LayerNorm(self.d_model)
129          if layer_type == "attn":
130              self.mixer = SlidingWindowAttention(self.d_model, self.n_heads, self.
                     window_size)
131          else:
132              self.mixer = MultiHeadFourier(self.d_model, self.n_heads)
133          self.ln_2 = nn.LayerNorm(self.d_model)
134          self.pos_ffn = MLP(self.d_model, self.intermediate_size)
135
136      def forward(self, x):
137          x = x + self.mixer(self.ln_1(x))
138          x = x + self.pos_ffn(self.ln_2(x))
139          return x
140
141
142  class CaracalForCausalLM(nn.Module):
143      def __init__(self, d_model, n_layers, n_heads, vocab_size, intermediate_size,
              attn_layers=(), window_size=256, **kwargs):
144          super().__init__()
145          self.d_model = d_model
146          self.n_layers = n_layers
147          self.n_heads = n_heads
148          self.vocab_size = vocab_size
149          self.intermediate_size = intermediate_size
150          self.window_size = window_size
151          self.attn_layers_set = set(attn_layers)
152          self.wte = nn.Embedding(self.vocab_size, self.d_model)
153          self.h = nn.ModuleList()
154          for i in range(n_layers):
155              layer_type = "attn" if i in self.attn_layers_set else "fft"
156              block = Block(
157                  d_model=self.d_model,
158                  n_heads=self.n_heads,
159                  intermediate_size=self.intermediate_size,
160                  window_size=window_size,
161                  layer_type=layer_type
162              )
163              self.h.append(block)
164          self.ln_f = nn.LayerNorm(self.d_model)
165          self.lm_head = nn.Linear(in_features=self.d_model, out_features=self.vocab_size,
                 bias=False)
166          self.wte.weight = self.lm_head.weight
167          self.apply(self._init_weights)
168
169      def forward(self, x):
170          x = self.wte(x)
171          for layer in self.h:
172              x = layer(x)
173          x = self.ln_f(x)
174          logits = self.lm_head(x)
175          return logits
176
177      def _init_weights(self, module):
178          if isinstance(module, (nn.Linear, nn.Conv1d)):
179              std = 0.02
180              if hasattr(module, 'NEED_SCALE_INIT'):
```

```
181            std *= (2 * self.n_layers) ** -0.5
182        torch.nn.init.normal_(module.weight, mean=0.0, std=std)
183        if module.bias is not None:
184            torch.nn.init.zeros_(module.bias)
185    elif isinstance(module, nn.Embedding):
186        torch.nn.init.normal_(module.weight, mean=0.0, std=0.02)
187
188
189 class Caracal(nn.Module):
190    def __init__(self, d_model, n_layers, n_heads, vocab_size, attn_layers=(),
           window_size=256, **kwargs):
191        super().__init__()
192        ffn_dim = int(2 * d_model * 4 / 3)
193        intermediate_size = (ffn_dim + 127) // 128 * 128
194        self.model = CaracalForCausalLM(d_model, n_layers, n_heads, vocab_size,
               intermediate_size, attn_layers, window_size)
195
196    def forward(self, x, targets=None, **kwargs):
197        logits = self.model(x)
198        loss = None
199        if targets is not None:
200            loss = F.cross_entropy(logits.view(-1, logits.shape[-1]), targets.view(-1))
201        return logits, loss
```

*Listing 1.* Caracal Implementation in PyTorch

## F. Limitations and Future Work

While our empirical results demonstrate the efficacy of **Caracal** across various scales, several directions remain for further investigation. First, our experiments validate the architecture up to current parameters; extensive large-scale pre-training is still required to verify its potential for trillion-parameter frontier models. Second, as discussed in Section 5.2.2, a "resolution gap" exists in fine-grained information retrieval tasks where **Caracal** currently trails behind dense attention-based models. We plan to explore architectural refinements, such as multi-scale gating or hierarchical mixing, to enhance localized retrieval capabilities while preserving sub-quadratic complexity. Finally, although **Caracal** inherently supports efficient incremental inference through a fixed-size rolling buffer (state), specialized optimization algorithms—analogous to the KV-cache management in Transformers—are still needed to fully maximize its inference throughput in production environments.

## G. Statement on the Use of Large Language Models

Throughout the preparation of this manuscript, we utilized a large language model (LLM) as a general-purpose assistant. The LLM's role was primarily that of a collaborative tool for language refinement, structural organization, and formatting, under the direct guidance and critical supervision of the human authors.

The specific uses of the LLM in the writing process include:

- **Language Refinement:** The LLM assisted in iteratively refining sentence structure, word choice, and overall tone to align with standard academic English. This process involved numerous cycles of author-led prompting, critical review, and editing to ensure the final text accurately reflected our intended meaning and technical nuances.

- **Figure, Table, Pseudocode and Citations Formatting:** It was used to generate LaTex code to insert figures and tables uploaded by authors. It was also used to convert a PyTorch code implementation of our MHF module into a LaTeX pseudocode algorithm for clarity. Besides, it helped with formatting citations in BibTeX and resolving LaTeX typesetting queries.

It is important to clarify that all core research ideas, the **Caracal** architecture design, the conceptual analyses, and the experimental framework are the original intellectual contributions of the human authors. The LLM did not contribute to the research ideation. The authors retained full intellectual control throughout the process, directed all revisions, and assume complete responsibility for the scientific validity and final content of this manuscript.

