# OpenReview forum: "Caracal: Causal Architecture via Spectral Mixing"
_ICML.cc/2026/Conference — ICML 2026 regular_

### Official Review · Reviewer_WtFy · 2026-03-11

**Soundness:** 2
**Presentation:** 2
**Significance:** 3
**Originality:** 2
**Overall Recommendation:** 4
**Confidence:** 3

**Summary:**

This paper introduces a causal FFT-based architecture designed to improve the efficiency of Transformer-style autoregressive modeling. Specifically, the method replaces the masked multi-head self-attention module with a multi-head Fourier module, and employs a pad-FFT-multiply-iFFT-truncate procedure to realize causal sequence mixing. Through this design, the model is intended to capture autoregressive dependencies without requiring explicit positional encodings, while achieving a computational complexity of $\mathcal{O}(L \log L)$, compared with the $\mathcal{O}(L^2)$ cost of standard self-attention. The proposed architecture is evaluated on a range of language modeling tasks to assess its effectiveness and efficiency.

**Compliance With Llm Reviewing Policy:**

Affirmed.

**Final Justification:**

After reading the rebuttal, I think the authors have addressed my main concerns satisfactorily and have responded seriously to both my questions and those of other reviewers. Although I still have some uncertainty about the strength of the novelty and superiority claims, I no longer view these issues as strong enough to outweigh the paper’s overall merits. I am willing to raise my score to 4.

**Key Questions For Authors:**

- In Section 3.1, the paper states that a small number of attention layers are retained and restricted to a sliding-window mechanism in order to capture local dependencies without quadratic cost. However, these sliding-window attention layers are neither described in sufficient detail in Section 3 nor illustrated in Figure 1, which makes the overall architecture difficult to parse. It would be helpful for the authors to clearly specify where these layers are inserted, how many such layers are used in practice, what window size is adopted, and whether these design choices are fixed across all experiments.
- In Section 3.2, Step 2 states that Layer Normalization is first applied after the causal pre-convolution, yielding $x_{norm} = LayerNorm(\tilde{x})$. However, this normalization step is not clearly aligned with the diagram in Figure 1, where the placement of LayerNorm relative to the Pre_Conv, $W_V$ and $W_{G1}$ modules is visually ambiguous. The paper would benefit from a clearer explanation of the exact placement of this normalization layer in both the text and the figure.
- The presentation of the multi-head Fourier mechanism is currently too implicit. The paper should explain more clearly how the heads are formed, how FFT/iFFT are applied across heads, whether frequency mixing is performed independently within each head, and how the outputs of different heads are aggregated before the final projection $W_O$.
- The paper claims that explicit positional encodings are unnecessary because the MHF module inherently captures global positional information, and because the pre-convolution injects local inductive bias. However, this claim is stronger than what is directly demonstrated in the current experiments. It would be valuable to include targeted ablations that isolate the contribution of removing positional encodings, for example by comparing against the same architecture with standard positional encodings added back.

**Limitations:**

Yes.

**Strengths And Weaknesses:**

Strengths:
- The paper addresses an important problem: replacing quadratic self-attention with a sub-quadratic mechanism for autoregressive modeling without relying on heavy hardware-specific optimizations. This is a meaningful and timely research direction.
- The proposed architecture is conceptually clean. The MHF block combines causal pre-convolution, content/gate branches, FFT-domain multiplication, iFFT, and truncation in a relatively simple and modular design.


Weaknesses:
- The methodology is not presented clearly enough for several core technical points. In particular, the relationship between the claimed kernel form $\alpha_{tj} = k_{t-j}(x)$ and the gate stream $x_g$ remains implicit, making it difficult to precisely interpret the proposed “context-aware” weighting mechanism from the main text alone.
- The causality argument is under-explained. The paper states that the “pad-FFT-multiply-iFFT-truncate” pipeline is equivalent to causal convolution, but does not make this equivalence sufficiently transparent and explicitly explained in Section 3, even though it is central to the method.
- The empirical results establish competitiveness more than superiority. Table 2 shows that Caracal is broadly competitive with Llama, Mamba, and Jamba, but not consistently better, which weakens the experimental soundness of the proposed method.
- The long-context results are not especially strong. In Table 4, Llama maintains a clear advantage on retrieval/extraction-oriented tasks such as SWDE and FDA, which weakens the claim of a broadly preferable long-context architecture.
- The novelty is not fully convincing. The paper’s main causality construction is a standard FFT realization of linear causal convolution via padding and truncation, and the effective adaptive kernel is generated by the gate stream $x_g$. Consequently, the proposed MHF block seems better understood as an input-conditioned causal convolution implemented in the frequency domain than as a genuinely new causal FFT mechanism.

---

> ### Author Rebuttal · Authors · 2026-03-31
>
> We thank the reviewer for the feedback and respond to your concerns.
>
> **W1: Relationship Between Kernel and Gate Stream**
>
> Our FFT pipeline uses **channel-wise causal convolution**. Gate stream $x_g \\in \\mathbb{R}^{L \\times d_{model}}$ (Section 3.2) provides **time-domain dynamic filter coefficients**. For channel $j \\in [\\{0, \\dots, d_{model}-1\\}]$, the sequence $x_g[:, j]$ defines the kernel $k^{(j)}(x)$. Specifically: $k^{(j)}_i(x) = x_g[i, j]$
>
> where $i$ is the temporal index ($i \\in [\\{0, \\dots, L-1\\}]$,). Because $x_g$ is dynamically generated from inputs $x$ via gate branch, each channel's convolutional kernel is **data-dependent**. We will clarify it in revision.
>
> **W2: Causality and Padding Logic**
>
> Due to the 5000-character limit, we outline our logic here; the full proof will be updated in the Appendix. The "pad-FFT-multiply-iFFT-truncate" pipeline realizes causal convolution via polynomial duality:
>
> 1. **Duality**: A length-$L$ sequence $v$ defines polynomial $P_v(z) = \\sum_{i=0}^{L-1} v_i z^i$, represented in **Coefficient Form** ($v$ itself) or **Point-Value Form** (evaluation $P_v(z)$ at $N$ roots of unity).
> 2. **FFT Shortcut**: Linear convolution matches the coefficients of $P_{out}(z) = P_v(z) P_k(z)$. FFTs bypass direct multiplication ($O(L^2)$) by converting $v$, $k$ to **Point-Value Form**, multiplying element-wise, and interpolating back to **Coefficient Form** via iFFT ($O(L \\log L)$).
> 3. **Causality & Padding**: FFTs are **circular**. Without padding ($N=L$), $z^L$ wraps to $z^0$. In $(v_0+v_1 z)(k_0+k_1 z)$, "future" $v_1 k_1 z^2$ wraps to $z^0$, contaminating "past" $out_0$ and violating causality.
> 4. **Truncation**: Padding $N \\ge 2L-1$ (we use $2L$) prevents wrap-around in causal window. Truncating the $2L$ result back to $L$ recovers causal coefficients where each $out_i$ depends only on $v$ and $k$ with indices $\\le i$.
>
> **W3: Superiority of Results**
>
> We respectfully disagree regarding our results' superiority.
>
> * **Superior Pareto Frontier:** Parity with highly optimized models like Llama plus a **3x throughput leap (Figure 2 and Section 5.3)** represents a superior Pareto frontier, not merely a "competitive" result.
> * **New Design Space:** Caracal is highly modular. Its components like gate branch and pre-convolution filters can be further innovated. By publishing these results, we offer a **flexible foundation for the community to further innovate** sequence mixing.
>
> **W4: Long-Context Retrieval and Extraction Performance**
>
> Please refer to response to **Reviewer dvUk (W1): Table 4 Performance (SWDE/FDA)**.
>
> **W5: Novelty**
>
> Our novelty lies in the **architectural synthesis of a scalable Transformer alternative**. While "pad-FFT-multiply-iFFT-truncate" is known for linear convolution, integrating it as a **trainable, data-dependent sequence mixer** in autoregressive models is novel.
>
> The reviewer labels MHF as "input-conditioned causal convolution." But this is precisely where the novelty lies compared to existing models:
> * **Vs. SSMs/Hyena:** Unlike Hyena’s data-independent filters, Caracal generates global mixing kernels dynamically.
> * **Vs. Attention:** Caracal matches attention's data-dependency but with $O(L \\log L)$ complexity in the frequency domain.
>
> AI innovation often involves repurposing math to solve bottlenecks (e.g., Diffusion Models via thermodynamics); MHF similarly contributes by bridging spectral efficiency with high-capacity sequence modeling.
>
> **Key Q1: Details of SWA Layers**
>
> Section 3.1 is intentionally modular as SWA placement is a flexible hyperparameter. Specific configurations reside in **Section 5.1 (Line 320-321)**: one SWA layer follows every two MHF layers (2:1 ratio) with window size of 256. We also evaluated varied positions in our Parameter Study **(Section 5.4, Table 5)**.
>
> **Key Q2: Placement of LayerNorm**
>
> We apologize for Figure 1's ambiguity. There are actually **two distinct LN layers**. **Block-level Pre-LN:** The **Pre-LN** before the `mixer` (MHF) as standard Transformer (shown in Figure 1). **Internal MHF-LN:** The **LN layer** immediately after the **pre-conv** (described in Section 3.2, Step 2) inside the MHF. We will update Figure 1 to show both of them in the revise paper.
>
> **Key Q3: Multi-Head of MHF**
>
> Setting the `pre_conv` with `groups = n_heads` **partitions $d_{model}$ channels into $n_{heads}$ independent sets**. Each head's convolution mixes information across its $d_{head}$ channels **without information exchange between different heads** during this stage. Following the pre-convolution and learnable projections, the **subsequent operations (FFT, Point-wise Interaction and iFFT) are strictly independent across channels**. There is no mixing across heads, nor across channels within a head. Finally, $n_{heads}$ outputs are concatenated and passed throught $W_O$ as standard Transformer. Details will be added to Appendix.
>
> **Key Q4: PE Ablation**
>
> Please refer to response to **Reviewer EKvP (W3 & W4)**.

---

> > ### Author Rebuttal · Reviewer_WtFy · 2026-04-02
> >
> > After reading the rebuttal, I think the authors have addressed my main concerns satisfactorily and have responded seriously to both my questions and those of other reviewers. Although I still have some uncertainty about the strength of the novelty and superiority claims, I no longer view these issues as strong enough to outweigh the paper’s overall merits. I am willing to raise my score to 4.

---

> > > ### Author Response · Authors · 2026-04-02
> > >
> > > Thank you very much for acknowledging our rebuttal and increasing the score to 4. We appreciate your professional and thorough evaluation. We will carefully incorporate your suggestions regarding the claims of novelty and superiority into the revised manuscript.

---

### Official Review · Reviewer_dY9c · 2026-03-13

**Soundness:** 2
**Presentation:** 1
**Significance:** 1
**Originality:** 1
**Overall Recommendation:** 3
**Confidence:** 4

**Summary:**

This paper proposes a replacement for standard causal O(N^2) attention mechanism with Fast Fourier Transform (FFT) based O(N logN) causal attention with standard operators unlike hardware-specific solutions. They pretrain with different scales (all less than 1B parameters) and types (LLAMA and State Space Models (SSMs)) of models on Lambada dataset.

**Compliance With Llm Reviewing Policy:**

Affirmed.

**Final Justification:**

As indicated my rebuttal acknowledgement, the rebuttal seems factual, and partially satisfactory.

**Key Questions For Authors:**

- Is it possible to train Caracal with the bfloat16 precision without encountering numerical problems?
- Can you provide the training time and throughput for training?
- Can you provide a result for standard attention using FlashAttention for LLAMA experiments?
- You need to correct the notation, N sometimes becomes batch size (line 637) and sequence length.
- Can you elaborate the interpretability aspect of frequency domain operations in MHF?

**Limitations:**

Yes

**Strengths And Weaknesses:**

Soundness: (2)

The Multi-Head Fourier (MHF) derivations are technically correct and well explained. However the practical sides and possible problems (utilization of the FFT kernel in PyTorch and possible numerical problems while mixed precision training with bfloat16) are not studied in detail. As for the experiments, only a single training setting is considered. Trained architectures are relatively small. Unlike the standard attention, MHF output is unbounded and this may cause instabilities during training a larger scale model with half precision.

Presentation: (1)

The paper is easy to follow but there is an inconsistency in the notation. Complexities are given in terms of L in the first pages and later it becomes in terms of N and in the end again it is in terms of L. Also The differences between the Hyena method (the most similar method) are not explained in detail.


Significance: (1)

The significance is limited. Only a single domain is addressed with small scale models.It is unclear how broadly the MHF would transfer to a larger and more realistic setting. The practical utility is also unclear since the FFT kernels are not optimized like Matrix Multiplication kernels. Torch.fft.rfft is not supported for bloat16. It also does not offer any KV cache mechanism during the inference and this could be a bottleneck in autoregressive generation settings.

 Originality: (1)

Although the paper does not acknowledge the connection between Hyena, it is quite related. Spectral mixing is not new and studied in many papers (FNet, GF-Net, AFNO) as an alternative to standard attention and only the causality is introduced.. Also the learned structure is unclear and less intuitive than the classical attention mechanism.

References:


Hyena: Poli, M., Massaroli, S., Nguyen, E., Fu, D., Dao, T., Baccus, S., Bengio, Y., Ermon, S., and Ré, C. Hyena Hierarchy: Towards larger convolutional language models. In International Conference on Machine Learning (ICML), volume 202, pp. 28043–28078, 2023.

FNet: Lee-Thorp, J., Ainslie, J., Eckstein, I., and Ontanon, S. FNet: Mixing tokens with Fourier transforms. In Proceedings of Conference of the North American Chapter of the Association for Computational Linguistics (NAACL), pp. 4296–4313, 2022.

GF-Net: Rao, Y., Zhao, W., Zhu, Z., Lu, J., and Zhou, J. Global filter networks for image classification. In Advances in Neural Information Processing Systems (NeurIPS), pp. 980–993,
2021.

AFNO: Guibas, J., Mardani, M., Li, Z., Tao, A., Anandkumar, A., and Catanzaro, B. Adaptive Fourier neural operators: Efficient token mixers for Transformers. arXiv preprint arXiv:2111.13587, 2022.

---

> ### Author Rebuttal · Authors · 2026-03-30
>
> We must address several fundamental factual misunderstandings in the review summary and comments, which suggest that key sections of our manuscript were overlooked.
>
> **1. Factual Corrections on Summary**
> * **Training Data:** The reviewer states we trained on the **Lambada** dataset. This is incorrect. As stated in **Section 5.1 (Line 324)**, we trained all models on **FineWeb**; Lambada was used strictly as an evaluation benchmark.
> * **Core Mechanism:** The reviewer describes Caracal as a "Fast Fourier Transform (FFT) based causal attention." We clarify that Caracal is **not** an attention mechanism. It is a **causal convolution-based sequence mixer** that entirely removes the attention bottleneck.
>
> **2. Soundness and Experimental Scale**
> * **Experimental Diversity:** The claim that "only a single training setting is considered" is contradicted by our results. We trained **four model scales** (63M to 724M parameters) to demonstrate Scaling Laws **(Table 2)**, trained a model with **longer context (Table 3)**, and conducted **extensive parameter studies** on MHF/SWA hybrid ratios **(Table 5)**.
> * **Numerical Stability:** Current library support for `torch.fft` in `bfloat16` is limited. We address this by casting inputs to `float32` within the module.
> * **Training Dynamics:** All MHF inputs and outputs are processed via LayerNorm. We monitored gradient norms across all scales and observed stable, consistent behavior without any sign of training instability or "unbounded" output issues.
>
> **3. Presentation and Citation**
> * **Notation ($L$ vs. $N$):** We define $N=2L$ in **Line 140** for causality padding in the main paper. In Line 637, $(N, C, L)$ refers to the standard library format for 1D convolutions, which we explicitly clarify in the text. We will unify this to $B$ in revision.
> * **Distinction from Hyena:** Detailed in **Line 60-68** of our paper, Hyena uses position-based (data-independent) filters, while Caracal employs content-based (data-dependent) filters generated dynamically from input tokens $x$. This content-awareness is a fundamental departure from Hyena.
>
> **4. Significance and Scalability**
> * **Model Scale:** Our experiments across four scales (up to 724M) provide strong empirical evidence that Caracal’s advantages follow clear **Scaling Laws** (Section 5.2) and are architecturally robust beyond "tiny" settings. While 724M is the limit of our current academic hardware (8x RTX 3090s), the performance trend is clear and competitive.
> * **FFT Operator Efficiency:** While FFT kernels lack specialized low-level optimization like FlashAttention, Caracal already outperforms optimized baselines (Section 5.3). This **underscores the inherent architectural efficiency** of our designs.
> * **Incremental Inference:** As a causal convolution, Caracal naturally supports **incremental inference** by maintaining a fixed-size **rolling buffer (state)**. We focus on the architectural foundation in this paper, leaving specific buffer engineering for future work.
>
> **5. Originality and Conceptual Intuition**
> * **Connection to Hyena:** Again, we dedicated a detailed discussion in **Line 60-68** to distinguish Caracal from Hyena.
> * **Evolution of Spectral Mixing:** Efficient causality was the fundamental "bottleneck" previously barring spectral methods from **generative/autoregressive** use (Section 4.2). Besides, unlike static works (FNet, AFNO), Caracal’s **dynamic, data-dependent mixing** enables the flexible, content-aware reasoning required for modern LLMs.
> * **Architectural Intuition:** **Section 4 (Conceptual Analysis)** demonstrates that both attention and FFT are fundamentally token-mixing mechanisms via **Weighted Sums** (Eqn. 1 & 2). Caracal bridges attention's data-dependency with FFT's efficiency, providing a clear "first-principles" theoretical grounding for replacing attention with MHF.
> * **Clarity of Structure:** The architecture is detailed in **Section 3**, pseudocode in the **Appendix** and source code in the **Supplemental Material**.
>
> **6. Key Questions**
> * **Q1 (bfloat16):** We use `float32` casting for FFT operations to circumvent library limitations, which does not hinder our superior training throughput.
> * **Q2 (Throughput):** Training time and throughput are extensively detailed in **Section 5.3 (Efficiency and Speed)** and **Figure 2**, occupying half a page of the paper.
> * **Q3 (Llama Baseline):** Our Llama tests use the **transformers** library's optimized **SDPA (Scaled Dot Product Attention)** implementation. Internal logging confirms the model initializes via the sdpa path on our hardware, ensuring a high-performance.
> * **Q4 (Notation):** We will further unify the notation in the final version to prevent any ambiguity.
> * **Q5 (Interpretability):** MHF acts as a **dynamic frequency-domain equalizer**, where heads learn to amplify or suppress specific frequencies based on input content—a mathematically grounded alternative to heuristic attention weights.

---

> > ### Author Rebuttal · Reviewer_dY9c · 2026-04-03
> >
> > The rebuttal is factual, and I am partially safisfied with the reviewers answers.
> >
> > I will increase my score to 3.
> >
> > One unaddressed question is the following, remaining from the original review. Do you have results with the FlashAttention?

---

> > > ### Author Response · Authors · 2026-04-04
> > >
> > > Thank you for your feedback and for acknowledging our efforts by increasing the score. Regarding your remaining question about **FlashAttention**, we would like to clarify that all of our Llama experiments already utilized FlashAttention-2 via the default SDPA path. As mentioned in our previous rebuttal, internal logging confirms that our hardware (NVIDIA RTX 3090) initializes the attention via the `sdpa` path. In modern deep learning stacks (PyTorch 2.0+), **SDPA (Scaled Dot Product Attention) is not a separate algorithm, but a kernel dispatcher** that automatically invokes FlashAttention when appropriate. We provide the following evidence from our environment:
> > >
> > > * **Evidence 1 (Integration):** In the `transformers` library (v$4.57.0$), the `sdpa_attention_forward` function (found in `transformers/integrations/sdpa_attention.py`) is the default implementation for Llama, which directly invokes `torch.nn.functional.scaled_dot_product_attention`.
> > > * **Evidence 2 (PyTorch Backend):** The official PyTorch documentation (and source code in `torch/nn/functional.py`) explicitly states that SDPA automatically selects from three backends, with **FlashAttention-2** being the first and preferred implementation for CUDA devices (like our RTX 3090).
> > > * **Evidence 3 (Empirical Profiling):** We conducted a targeted trace of the core SDPA operation used by our Llama implementation. The profiling results confirm the mandatory invocation of the dedicated FlashAttention-2 kernel. A summary of the CUDA kernel trace is shown below:
> > >
> > > | Name | Self CUDA % | Self CUDA | # of Calls |
> > > | :--- | :--- | :--- | :--- |
> > > | void pytorch\_flash::flash\_fwd\_kernel<... | 100.00% | 14.208us | 1 |
> > > | cudaLaunchKernel | 0.00% | 0.000us | 1 |
> > >
> > > The kernel **pytorch\_flash::flash\_fwd\_kernel** is the **official FlashAttention-2 forward kernel** integrated into PyTorch.
> > >
> > > We hope this clarifies your remaining concern.

---

### Official Review · Reviewer_dvUk · 2026-03-13

**Soundness:** 3
**Presentation:** 4
**Significance:** 3
**Originality:** 2
**Overall Recommendation:** 4
**Confidence:** 3

**Summary:**

The paper introduces Caracal, a causal language model that replaces most of its attention layers with a multi-head Fourier module which reduces the computational complexity from $L^2$ to $L logL$, offering an efficient alternative architecture. Only a few layers retain a sliding window attention mechanism to capture local dependencies. The MHF module uses an FFT based mixer, removes the need for positional embeddings and does not need any hardware aware implementations.

**Compliance With Llm Reviewing Policy:**

Affirmed.

**Final Justification:**

I thank the authors for their detailed rebuttal. Most of my concerns were addressed. Although the current empirical results are not very convincing in long context performance and novelty remains limited, I find the core architectural idea interesting and potentially useful. Considering the authors’ willingness to soften some of the stronger claims regarding long context, I will increase my score.

**Key Questions For Authors:**

1. Could you provide a peak memory comparison of these models at different context lenghts as it is another important factor affecting scalibility?
2. It is written that removing PE is “theoretically advantageous for length extrapolation,” and even talks about “infinite context capability,” but important tests to measure this capability like length extrapolation, needle-in-a-haystack are not available. The current long-context evaluation is too narrow for the strength of the claims. Could you add more comprehensive tests to further show the advantages of removing the PE?

**Limitations:**

Yes.

**Strengths And Weaknesses:**

**Strengths:**
- The motivation of the paper is very clear and important.
- The presentation is easy to follow and understand.
- The hybridization ablation provides informative intuition.

**Weaknesses:**
1.  Caracal is presented as a competitive and efficient long sequence model, however the experimental results do not support this strong claim. In Table 2, although it shows competitive performance most of the time, the context lenght is 512, which is not ‘long’. In Table 4, which is the most relevant table for the intended use case, Caracal idoes not perform strongly enough to be considered competitive.
2. Given the existing literature on FFT based architectures, the novelty appears limited. A clear distinction from prior work and emprical comparison with these similar methods could strengthen the positioning of the paper.
3. The SSM portion of the related work is very limited. Never models such as some of the baselines in Table 3 are not discussed or cited.
4. Ablations on key architectural choices are missing. For instance, the claims regarding impact of initial causal convolution, the two stage desing of $W_{G1}$ and $W_{G2}$, PE being unnecessary, pure MHF vs hybrid should be emprically discussed. Without these ablations, it is hard to understand how each component contributes.
5. Caracal is argued to be more portable and simpler than SSM based approaches since it relies on standard FFT operators rather than specialized kernels. Although this is true with respect to some earlier implementations, more recent SSD / Mamba-2–style formulations have been using classical matmul/einsum implementations. This weakens the method’s advantages over some SSMs.

---

> ### Author Rebuttal · Authors · 2026-03-31
>
> We thank the reviewer for their comments and respond below.
>
> **W1. Context Length in Training and Its Performance**
>
> **Context in Training**: Due to hardware constraints (8x RTX 3090), 512 was the maximum feasible length for the L-size model. However, **Table 3 (4096 context, M size)** demonstrates Caracal's advantage as length increases.
>
> **Table 4 Performance (SWDE/FDA)**: **Llama** uses pointwise matching for near-perfect fine-grained retrieval. Like **Mamba**, Caracal favors efficiency, yielding lower "resolution" for exact extraction. Thus, Caracal offers a superior Pareto frontier for global modeling, though not yet a full replacement for localized retrieval tasks. We will refine our 'long context' claim and explore multi-scale gating to bridge this gap while remaining sub-quadratic in future work.
>
> **W2. Distinction from Prior FFT Models**
>
> Unlike non-causal spectral models (**FNet** and **AFNO**), Caracal resolves the **Causality Dilemma (Section 4.2, Line 231-274)**. Compared to autoregressive-capable spectral models: 1) **SPECTRE** relies on fixed sliding windows, fragmenting long-range dependencies, while Caracal maintains a strictly global receptive field. 2) **FlashButterfly** uses static kernels, while Caracal generates filters dynamically from input, providing superior adaptability.
>
> **W3. SSM Related Work**
>
> We have updated the Related Work section to include these important recent advancements: **RetNet** (Sun et al., 2023), **GLA** (Yang et al., 2024), **DeltaNet** (Yang et al., 2024), **Gated DeltaNet** (Yang et al., 2025), **TTT** (Sun et al., 2024). We cannot put details of them here due to the 5000-character limit of this rebuttal.
>
> **W4. Ablation Study**
>
> Please refer to **W3 & W4** in our response to **Reviewer EKvP**. These results will be included in the revised paper.
>
> **W5. Portability Compared to Modern SSMs**
>
> Despite modern SSMs adopting matmul/einsum, Caracal offers superior **Structural Clarity** and **Easier Implementation**:
>
> **Flexibility & Design Logic**: SSD-style models involve a **high conceptual barrier**, requiring deep expertise in complex block-partitioning and state-space duality. This **black-box nature** makes it challenging to modify internal logic without breaking the underlying mathematical framework. In contrast, by viewing Caracal as a causal convolution, researchers can innovate on components like the pre-convolution or the composition of $W_{G1}$ and $W_{G2}$ without risking mathematical misalignment or efficiency breakdowns.
>
> **Deployment Simplicity**: While modern SSMs can use standard operators, they still require **manual tuning** of block sizes and alignments to achieve peak performance. Caracal delivers **high speed by default** using standard, highly-optimized FFT libraries. This makes it more straightforward to maintain in diverse production environments.
>
> **Key Q1. Memory Scalability**
>
> Peak training memory with different context lengths (Tiny-size, 8x NVIDIA RTX 3090, `torch.cuda.max_memory_allocated()`) in MB:
>
> | Model | 256 | 512 | 1024 | 2048 | 4096 | 8192 |
> | :--- | :---: | :---: | :---: | :---: | :---: | :---: |
> | Llama | 1568.70 | 1909.97 | 2590.61 | 3979.16 | 6674.99 | 12112.68 |
> | Mamba | 1628.99 | 1881.37 | 2471.99 | 3655.09 | 6025.54 | 10758.83 |
> | Mamba2 | 1588.05 | 1834.02 | 2424.66 | 3605.00 | 5962.87 | 10684.01 |
> | Jamba | 1770.83 | 2154.49 | 2937.85 | 4519.32 | 7619.90 | 13988.52 |
> | Caracal | 1592.53 | 1897.09 | 2553.51 | 3880.19 | 6437.35 | 11619.61 |
>
> Llama remains competitive via optimized SDPA (Scaled Dot-Product Attention) despite quadratic complexity. Even so, Caracal consistently maintains a lower memory footprint than both Llama and Jamba.
>
> **Key Q2. Needle in a Haystack Test**
>
> The results of the NIAH test across several architectures are summarized below (Scores range from 1 to 10, where 10 is optimal):
>
> | Len(Depth%) | 200(10) | 200(50) | 200(90) | 500(10) | 500(50) | 500(90) | 800(10) | 800(50) | 800(90) | 1k(10) | 1k(50) | 1k(90) |
> | :--- | :---: | :---: | :---: | :---: | :---: | :---: | :---: | :---: | :---: | :---: | :---: | :---: |
> | Llama | 3 | 1 | 3 | 5 | 3 | 1 | 3 | 1 | 1 | 1 | 1 | 1 |
> | Mamba | 1 | 1 | 5 | 1 | 3 | 1 | 1 | 3 | 1 | 1 | 1 | 1 |
> | Mamba2 | 3 | 1 | 5 | 1 | 1 | 1 | 3 | 1 | 1 | 1 | 1 | 1 |
> | Jamba | 3 | 3 | 3 | 1 | 1 | 3 | 3 | 1 | 1 | 1 | 1 | 1 |
> | Caracal | 3 | 5 | 1 | 3 | 1 | 1 | 1 | 1 | 3 | 1 | 1 | 1 |
>
> At 1k length, all models (including attention-based Llama) hit the performance floor (score 1). This suggests that at our current parameter scale and training budget (limited by 8x RTX 3090s), NIAH becomes a non-discriminative benchmark as the bottleneck is resource-dependent rather than architectural. We will refine our "infinite context" claims to emphasize their theoretical basis (afforded by MHF's shift-invariance) while acknowledging these empirical limits.

---

> > ### Author Rebuttal · Reviewer_dvUk · 2026-04-02
> >
> > I thank the authors for their detailed rebuttal. Most of my concerns were addressed. Although the current empirical results are not very convincing in long context performance and novelty remains limited, I find the core architectural idea interesting and potentially useful. Considering the authors’ willingness to soften some of the stronger claims regarding long context, I will increase my score.

---

> > > ### Author Response · Authors · 2026-04-02
> > >
> > > Thank you very much for your feedback and for increasing the score to 4. We are encouraged that you found our core architectural idea interesting and potentially useful. We will refine the presentation of our long-context results in the revised manuscript to ensure a more objective and measured characterization of our method.

---

### Official Review · Reviewer_EKvP · 2026-03-24

**Soundness:** 3
**Presentation:** 3
**Significance:** 4
**Originality:** 4
**Overall Recommendation:** 5
**Confidence:** 4

**Summary:**

The main contribution of this work is a novel convolution-based sequence mixer algorithm which aims at replacing Transformer module and operates with reduced $O(L \log L)$ computational complexity. It includes a sequence-length-long convolution kernel with the weight being a function of inputs, rather than just some learnable parameter. That direct dependency on the input is an innovation I have not seen elsewhere in the literature.

Acknowledgement: I was a reviewer for this work at another conference. At that time I leant to reject this paper. But in this new version the authors addressed the majority of my feedback and significantly improved the paper, to the point that the most serious previous weaknesses and questions were remediated or answered. Now I'm leaning to accept, although there are still some concerns I would like to be resolved.

**Compliance With Llm Reviewing Policy:**

Affirmed.

**Final Justification:**

The rebuttal addressed my main concerns, please see "Rebuttal Acknowledgement" for detailed explanation.

**Key Questions For Authors:**

1. Did the speed comparisons in Figure 2 employ efficient low-level kernels for Llama (Flash-Attention) and for Mamba 1-2 or did you use plain PyTorch implementation for these models, too?

2. Also, did you use pure of hybrid Caracal variant for speed measurements? If hybrid, which configuration did you use and was FlashAttention for SWA incorporated?

3. Why did you choose to name the architecture as Caracal?

**Limitations:**

Yes.

**Strengths And Weaknesses:**

**Strengths**

* This work is highly relevant as it aims to address a pressing problem of algorithmic inefficiency currently dominating Transformer architecture on long sequences.

* The method is novel and sufficiently distinct from previous attempts to adopt long convolutions / Fourier Transform as a sequence mixer, of which I am aware.

* I’m really surprised about the empirical effectiveness of the MHF architecture, despite the lack of recency bias. If properly explained or ablated, this is a very important revelation.

* The validation results are competitive with Softmax attention, modern SSMs, and a Transformer-SSM hybrid (Jamba). The scope of evaluation is comprehensive and it includes language understanding and generation benchmarks, recall-heavy tasks and speed comparisons with other sub-quadratic architectures and Transformer.

* The code is provided, its implementation is correct and aligns with the explanations from the paper. It brings a great advantage in terms of reproducibility and future follow-ups/ iterations on this work.

* A major and very appealing advantage is that the architecture doesn’t require hardware-specific custom CUDA kernels and is formulated on several dozens of lines of code in PyTorch.


**Weaknesses**

**1.**

Lines 270-274: It was hard to convince myself that casualty is not broken until I recalled that the architecture’s sequence mixer is, in fact, a data-dependent convolution, and the whole chain of computations “pad-FFT-multiply-iFFT-truncate” is just an efficient way to compute it. I would advise to explicitly state that FFT is a convenience tool and to showcase the real underlying target calculation:

$g(t) = f_1(x_t), v(t) = f_2(x_t), out(t)=\sum_{k=0}^{t} v(k) g(t-k)$

**2.**

I’m still surprised about the empirical effectiveness of the above formulation. It means that for each subsequence ending with timestamp t, the corresponding subsequence $\{g(0), …, g(t)\}$ is reversed before element-wise multiplication with $\{v(0), …, v(t)\}$, and element $g(t)$ always interacts with $v(0)$, $g(t-1)$ with $v(1)$ and so on. Basically, the opposite elements of the subsequences $v$ and $g$ interact multiplicatively while elements close in time (like $v(t)$ and $g(t-1)$) don’t. It breaks the recency bias, inherent to both global and sliding window attention, as well as SSMs.

I still believe this trait of the architecture requires further exploration and analysis of why it works competitively this way. Please provide some reasoning for this phenomenon in the paper. Additionally, an ablation of what would happen if you replace convolution by cross-correlation: $out(t)=\sum_{k=0}^{t} v(k) g(k - t)$, or z-shifted cross-correlation: $out(t)=\sum_{k=0}^{t} v(k) g(k - t + z)$, could benefit the work and perhaps produce even better empirical results. Please note that both variants I formulated don’t break the causality.

Edit: I realized that the first suggested option $out(t)=\sum_{k=0}^{t} v(k) g(k - t)$ would be equivalent to $v(t)$, which is obviously redundant. But the second option $out(t)=\sum_{k=0}^{t} v(k) g(k - t + z)$ still seems viable.

**3.**

The MHF module uses short convolutions. From personal experience and existing body of literature (see e.g., Physics of Language Models: Part 4.1, https://arxiv.org/abs/2512.17351) I know that this architectural element can bring massive gains in performance. Thus, it would be interesting to examine how much of the performance of your architecture could be attributed to the short conv.

**4.**

I see a minor problem in evaluations on benchmarks in Table 3. Caracal is a *hybrid* MHF + sliding-window attention architecture, while all other models, except for Transformer++, are *pure* linear RNNs / SSMs. Incorporation of SWA layers usually brings notable performance boost to such architectures (see e.g., Gated Delta Net paper, https://arxiv.org/abs/2412.06464). Could you additionally train and evaluate the pure-MHF Caracal variant?

**5.**

There is no proof of Theorem 1 on lines 686-687.

**6.**

Minor typos/ presentation concerns

* Lines 89-92: it would be judicious to note somewhere in the text that Caracal has greater computational complexity than SSMs ($O(L \log L)$ vs $O(L)$).

* Equation 3 depicts $W_{G_2}$ as a weight parameter, when in reality it's a convolution operator applied to whether $x_{norm}$ or to the output of SiLU (this begs the question: to what, exactly?).

* Please present a Table (perhaps in the Appendix) which records the underlying data from Figure 2, because the figure is too small and it’s hard to discern the differences in speed of different architectures.

* Another suggestion: you could include the code of `MultiHeadFourier` class as a listing in the Appendix for completeness and self-containedness.

---

> ### Author Rebuttal · Authors · 2026-03-31
>
> We thank the reviewer for continued guidance and respond your remaining concerns below.
>
> **W1 & W2. Statement and Explanation of underlying Convolution**
>
> The equation in your comment perfectly capture the MHF mechanism and we will add it to the paper. If we understand the suggested z-shifted cross-correlation correctly (let z=2 for simplicity), the sequence of operations would look like:
>
> $out(0)=v(0)*g(2)$
>
> $out(1)=v(0)*g(1)+v(1)*g(2)$
>
> $out(2)=v(0)*g(0)+v(1)*g(1)+v(2)*g(2)$
>
> $out(t)=v(t-2)*g(0)+v(t-1)*g(1)+v(t)*g(2)$
>
> This method ensures v(t) interacts with a fixed "anchor" in the kernel. However, this variant primarily uses a local window of the kernel (g(0) to g(z)). As the sequence progresses, the influence of early tokens naturally diminishes as they shift out of the kernel's active range.
>
> To fully leverage the "sequence-length-long" property of our dynamic kernel for long-range sequence mixing, we eventually opted for the standard causal convolution:
>
> $out(0)=\\mathbf{v(0)*g(0)}$
>
> $out(1)=v(0)*g(1)+\\mathbf{v(1)*g(0)}$
>
> $out(2)=v(0)*g(2)+v(1)*g(1)+\\mathbf{v(2)*g(0)}$
>
> ...
>
> This allows every token (like v(0)) to maintain continuous interaction with the kernel across the entire history. g(0) consistently interacts with different v(t) and learns the weight for the "current step", while g(dist) learns the decay or importance of "past tokens" at that specific distance.
>
> **W3 & W4. Ablation Study**
>
> Following your constructive suggestions, we evaluated the contributions of Short Convolutions (referred to as `pre_conv` in our paper and 'Initial Causal Convolution' by another reviewer, hereafter abbreviated as **PC**) and Sliding-Window Attention (**SWA**). We will train a Pure MHF model and add it to Table 3 (WIP). Besides, we also included additional tests on adding Positional Encoding (**PE**) or replacing the two-stage gated design ($W_{G1}, W_{G2}$) with a Single-Stage Linear Projection (**SSLP**). All the tests were performed on our L-size model:
>
> | Model Variant | Hella. | ARC-e | ARC-c | Wino. | BoolQ | PIQA | SIQA | LMB.acc | Avg. | LMB.ppl | SWDE | FDA |
> | :--- | :---: | :---: | :---: | :---: | :---: | :---: | :---: | :---: | :---: | :---: | :---: | :---: |
> | Full Model | 45.10 | 58.16 | 29.69 | 53.20 | 61.90 | 69.26 | 39.51 | 35.26 | 49.01 | 29.39 | 8.82| 1.91 |
> | w/o SWA | 42.58 | 57.53 | 30.55 | 52.72 | 61.83 | 69.80 | 40.12 | 30.62 | 48.22 | 54.54 | 8.64 | 1.81 |
> | w/o SWA & PC | 41.86 | 56.57 | 29.69 | 50.43 | 63.39 | 71.00 | 39.51 | 30.10 | 47.82 | 60.85 | 8.55 | 1.81 |
> | with PE | 44.21 | 57.49 | 30.46 | 52.80 | 64.01 | 67.03 | 40.53 | 35.01 | 48.94 | 30.89 | 8.82 | 1.91 |
> | SSLP | 42.25 | 56.90 | 29.86 | 49.72 | 65.41 | 69.42 | 39.10 | 31.75 | 48.05 | 55.98 | 8.64 | 1.72 |
>
> Findings: 1) SWA and PC provide crucial local priors, but core MHF remains competitive even without them. 2) Adding PE yields no significant gain, confirming MHF’s inherent positional awareness. 3) The two-stage gating beats SSLP in capturing complex semantics.
>
> **W5. Proof of Theorem 1**
>
> Due to the 5000-character limit, we summarize the proof logic here and will include the full derivation in Appendix.
>
> 1. **Attention**: For unit shift $r_t = v_{t-1}$, Attention matrix $A_{attn}$ must learn $N$ distinct query-key alignments across rows. Since each row is independent, the degrees of freedom required to specify this alignment across the full sequence length $N$ scale as $O(N^2)$ within the matrix structure.
>
> 2. **MHF**: MFH matrix $A_{mhf}$ is constrained to be Toeplitz ($A_{t,j} = k_{t-j}$). Representing the same shift only requires setting $k_1 = 1$, ensureing $\\alpha_{t, t-1} = 1$ for all $t$. This shift-invariant inductive bias reduces the required degrees of freedom to $O(1)$.
>
> **W6. Presentation concerns.**
>
> **Lines 89-92**: We will explicitly note that Caracal operates with $O(L \\log L)$ complexity due to FFT-based mixing, compared to the $O(L)$ complexity of typical SSMs.
>
> **Equation 3**: $W_{G2}$ is the convolution operator applied to the output of the SiLU activation within the gated branch.
>
> **Data & Code**: Numerical data for Figure 2 and the MHF code will be added to the Appendix.
>
> **Key Q1. Speed Benchmark Optimizations**
>
> All baselines used optimized kernels: `mamba_ssm` (Selective Scan/SSD) for Mamba/Mamba2 and `transformers` (flash-attention) for llama.
>
> **Key Q2. Model Configuration and SWA Optimization**
>
> Figure 2 used the **Hybrid (2:1)** with `flash_attn_func`, identical to our main experiments. Pure MHF Caracal is slightly slower. For context length from 256 to 8192 with other conditions the same, the total training time (s) of pure MHF Caracal is 35749, 35781, 36297, 36303, 37461, 37785. It increases training time by 2776s to 3714s relative to the hybrid model.
>
> **Key Q3. Name of Model**
>
> Caracal has sensitive ears (spectral processing) which serves as a metaphor for our MHF. Besides, it is a backronym derived from our paper's title: **C**ausal **Ar**chitecture vi**a** Spe**c**tr**al** Mixing.

---

> > ### Author Rebuttal · Reviewer_EKvP · 2026-04-02
> >
> > W1: I'm satisfied by the answer, please add the formula to the final version.
> >
> > W2: I'm satisfied as long as would also add these discussions to the paper. It's really interesting how strong the performance of your architecture is despite the apparent lack of proximity inductive bias like the one present in SSMs or SWA. It would benefit the paper if you explicitly include these derivations which show that alternative formulations cross-correlation and so called "z-shifted" cross-correlation are not viable options because they consider just a few tokens for each subsequence.
> >
> > W3: I'm a little confused by you mentioning "We will train a Pure MHF model and add it to Table 3 (WIP)". Isn't the option "w/o SWA & PC" is the same as Pure MHF? I think you call a "Pure MHF" the variant with MFH+ShortConv, but please clarify it. Otherwise, I'm satisfied.
> >
> > W4. I'm fully satisfied by the answer.
> >
> > W5, W6 I'm satisfied if you will include the full proof and clarifications in the final revision.
> >
> > Questions: Thank you for answering them. Please include the answers in the final version, and don't be uncomfortable that pure MHF is slower. I as said earlier, negative results can also be very important and thought-provoking for the community. Please also add throughput data for pure MHF data.
> >
> > ---
> >
> > **In conclusion**, my feedback has been resolved, and I'm happy to increase the rating to 5. I think it's a good work and vote for its acceptance on the condition that authors incorporate *all* their answers from rebuttal in the final version.

---

> > > ### Author Response · Authors · 2026-04-03
> > >
> > > We are deeply grateful for your continued engagement and for increasing the score to 5. Your suggestions have significantly helped us improve the depth of our work. We address your remaining points below:
> > >
> > > **W2 (Analysis of Inductive Bias & Alternatives)**: > Thank you for the insightful observation regarding the lack of traditional proximity bias. We agree that this is a unique trait of our architecture. We will explicitly include the theoretical derivations from our rebuttal in the revised manuscript to discuss why alternative formulations (like z-shifted cross-correlation) are less viable due to their limited local window. We will also add a brief discussion on how Caracal achieves its competitive performance by allowing for global interactions across the sequence, providing a more flexible alternative to fixed proximity priors.
> > >
> > > **W3 (Clarification of "Pure MHF")**: > We apologize for the confusion. To clarify: you are correct. We define **Pure MHF** as the **variant consisting of MHF with ShortConv (PC) integrated, but without SWA**. The "w/o SWA & PC" entry in our ablation table was strictly intended to isolate the specific contribution of the ShortConv layer itself (relative to "w/o SWA"). In the revised Table 3, the "Pure MHF" baseline will correspond to the "w/o SWA" configuration.
> > >
> > > **W4 (Table 3)**: > In addition to the ablation study listed in our rebuttal, we are finalizing the training of the Caracal model (M-size, 4096 context) without SWA. We will include its performance metrics in the revised Table 3.
> > >
> > > We will ensure all formulas, proofs, and discussions from this rebuttal are faithfully incorporated into the revised manuscript. This includes a transparent report of the results for the pure MHF variant, acknowledging its lower throughput relative to the SWA-integrated version. Thank you again for your strong support!

---

### Decision · Program_Chairs · 2026-04-30

**Decision:**

Accept (regular)

**Comment:**

This paper proposes a new subquadratic layer based on data-dependent causal convolutions. Empirical results are promising, showing parity with other popular sequence models such as Transformers and Mamba. Although reviewers initially raised some concerns, the rebuttal addressed most concerns and reviewers generally recommended acceptance. While some concerns remain around scalability, these are not fundamental in light of the paper proposing new architectural directions that require some time for community support. As such, I recommend acceptance.